



# Validating the Nernst–Planck transport model under reaction-driven flow conditions using RetroPy v1.0

Po-Wei Huang[1], Bernd Flemisch[2], Chao-Zhong Qin[3], Martin O. Saar[1,4], and Anozie Ebigbo[5]

[1]Geothermal Energy and Geofluids Group, Institute of Geophysics, Department of Earth Sciences, ETH Zurich, Zurich, Switzerland
[2]Institute for Modelling Hydraulic and Environmental Systems, University of Stuttgart, Stuttgart, Germany
[3]State Key Laboratory of Coal Mine Disaster Dynamics and Control, Chongqing University, Chongqing, China
[4]Department of Earth and Environmental Sciences, University of Minnesota, Minneapolis, USA
[5]Chair of Hydromechanics, Helmut Schmidt University, Hamburg, Germany

**Correspondence:** Po-Wei Huang(powei.huang@erdw.ethz.ch), Anozie Ebigbo(ebigbo@hsu-hh.de)

**Abstract.** Reactive transport processes in natural environments often involve many ionic species. The diffusivities of ionic species vary. Since assigning different diffusivities in the advection-diffusion equation leads to charge imbalance, a single diffusivity is usually used for all species. In this work, we apply the Nernst–Planck equation, which resolves unequal diffusivities of the species in an electroneutral manner, to model reactive transport. To demonstrate the advantages of the Nernst–Planck

model, we compare the simulation results of transport under reaction-driven flow conditions using the Nernst–Planck model with those of the commonly used single-diffusivity model. All simulations are also compared to well-defined experiments. Our results show that the Nernst–Planck model is valid and particularly relevant for modeling reactive transport processes with an intricate interplay among diffusion, reaction, electromigration, and density-driven convection.

## 1   Introduction

Reactive transport processes are fundamental in large-scale subsurface applications such as: geological hydrogen storage (Aftab et al., 2022; Hassanpouryouzband et al., 2022), geothermal energy extraction (Randolph and Saar, 2011; Fleming et al., 2020; Ezekiel et al., 2022), biogeochemical subsurface treatments (Carrera et al., 2022), contaminant remediation (Frizon et al., 2003; Reddy and Cameselle, 2009), recovery of minerals (Chang et al., 2021), and carbon sequestration and storage (Luhmann et al., 2013; Tutolo et al., 2014; Pogge von Strandmann et al., 2019; Grimm Lima et al., 2020; Ling et al., 2021). On a

smaller scale, reactive transport governs the processes of steel corrosion in reinforced concrete (Yu et al., 2020), cementation processes (Samson and Marchand, 2007; Cochepin et al., 2008), dendrite growth in microelectronics (Illés et al., 2022), and electrochemical reactors (Rivera et al., 2021). In all aforementioned settings, the reacting fluid consists of multiple ionic species and may be subject to an external electric field. In some applications, the electromigration of ionic species can play a significant role in the overall transport process. A typical model, which accounts for electromigration, is the Nernst–Planck equation.

Modeling electrochemical processes using the Nernst–Planck equation is usually coupled with the Poisson equation for the electrical potential, also known as the Poisson–Nernst–Planck (PNP) model. The PNP model has been applied in vari-





ous electrochemical applications (Jasielec, 2021). In geoscientific applications, benchmarks of the Nernst–Planck model have been proposed and verified considering charged clay surfaces (Tournassat and Steefel, 2021). Charged surfaces affect the water permeability of a porous material (in contrast to air permeability, which is not affected by electric charges) (Revil and

Leroy, 2004; Kwon et al., 2004b, a; Cheng and Milsch, 2020). The PNP model is utilized at the pore scale to determine such water permeabilities (Priya et al., 2021). Pore-scale experiments with external charge effects have also been conducted to investigate reactive transport processes (Sprocati et al., 2019; Sprocati and Rolle, 2022; Rolle et al., 2022; Izumoto et al., 2022). Rasouli et al. (2015) verified numerical software using electromigration benchmarks (Lichtner, 1994; Glaus et al., 2013) and multicomponent-electromigration experiments considering dispersion effects (Rolle et al., 2013). Finally, multicomponent

transport experiments in porous media confirm the validity of the Nernst–Planck model (Muniruzzaman et al., 2014; Muniruzzaman and Rolle, 2015, 2017; Rolle et al., 2018).

The above benchmark and validation studies focused on reactions and the mixing of fluids, subject to flow introduced by fluid pressure or electric fields. In this work, we propose another set of benchmark problems for numerical models, based on chemically driven convection experiments (Avnir and Kagan, 1984; Eckert and Grahn, 1999; Eckert et al., 2004), which focus

on how reaction and mixing of fluids influence fluid flow. The concept of reaction-driven flow is as follows: Consider reaction $A + B \rightarrow C$, where A and B are the reactants and C is the product. If A, B, and C have different material properties, for example, viscosity, diffusivity, and/or density, the reaction can induce fluid flow and hydrodynamic instabilities in such systems (De Wit, 2016, 2020). We are interested in aqueous systems, where ionic species exist, such that the Nernst–Planck model is applicable.

Zalts et al. (2008) performed experiments using an acid and a base to study chemically driven convection. Related exper-

iments and theories of the underlying mechanism have emerged: unequal species diffusivities (Citri et al., 1990; Almarcha et al., 2010, 2011; Trevelyan et al., 2011; Lemaigre et al., 2013; Kim, 2019; Jotkar et al., 2021), the same species diffusivity but with varying viscosity (Hejazi and Azaiez, 2013), concentration-dependent species diffusivities (Bratsun et al., 2015, 2017, 2021, 2022). Chemically driven convection has also been investigated using a weak acid and a weak base (Cherezov et al., 2018). Since the Nernst–Planck model is expected to be able to account for unequal species diffusivities and has the

characteristics of concentration-dependent diffusive fluxes, we examine the validity of the Nernst–Planck model by comparing our Nernst–Planck simulations with chemically driven experiments, conducted by others.

In addition, we present simulations of convective dissolution of $CO_2$, a process which is important in the underground storage of $CO_2$ (Singh et al., 2019). Thomas et al. (2016) performed experiments of convective dissolution of $CO_2$ into alkaline brines, including $LiOH$ and $NaOH$, and observed that different cations lead to varying onset times and fingering patterns. Thomas

et al. (2016) proposed that the inert cation, which does not participate in the acid-base reaction between the dissolved $CO_2$ and the brine, plays a major role in the development of convective instabilities. Since dissolved $CO_2$ forms charged and uncharged states ($CO_2(aq)$, $HCO_3^-$, $CO_3^{2-}$) when the pH changes as a consequence of neutralization reactions, it is natural to apply the Nernst–Planck model in such cases.

In the following sections, we first establish the fundamental equations of the conservation laws and the accompanying

numerical methods. We then provide details concerning the two reaction-driven flow experiments and a $CO_2$ dissolution experiment—all conducted by others. We compare the differences between the simulations of the single-diffusivity transport





model and the Nernst–Planck model, and elucidate the necessity of considering electromigration effects caused by differing diffusivities during reactive transport modeling.

## 2 Materials and Methods

### 2.1 Fundamental equations for reactive transport processes

In this section, we delineate the fundamental equations describing the reactive transport of aqueous species.

### 2.1.1 Mass balance of the fluid components

We consider a single-phase fluid composed of $N$ components, which can be ionic species, dissolved gas, or minerals. The single-phase fluid occupies a certain volume $V_f$, and the $i$th component of the fluid has the mass $m_i$. The mass concentration
of the $i$th component is defined as

$$\rho_i = \frac{m_i}{V_f}. \tag{1}$$

We use the term mass concentration to clarify that $\rho_i$ is not the density of the component. The fluid density, $\rho$, can be obtained by summing up the mass concentrations of the components, i.e.,

$$\rho = \sum_{i=1}^{N} \rho_i = \sum_{i=1}^{N} \frac{m_i}{V_f}. \tag{2}$$

In general, the fluid density is a function of fluid pressure, $p$, temperature, $T$, and fluid composition (i.e., the mass concentrations, $\rho_i$). The equation of state of the fluid density can be written as

$$\rho = \rho(p, T, \rho_i;\ i = 1, 2, ..., N). \tag{3}$$

The mass balance of each fluid component is described by the continuity equation,

$$\frac{\partial \rho_i}{\partial t} + \nabla \cdot (\rho_i \boldsymbol{v}_i) = Q_i, \tag{4}$$

where $\rho_i \boldsymbol{v}_i$ is the mass flux and $Q_i$ is the source/sink term introduced by chemical reactions. Since chemical reactions preserve fluid mass, we have $\sum_i Q_i = 0$. We express the mass flux of the $i$th component as

$$\rho_i \boldsymbol{v}_i = \rho_i \boldsymbol{u} + \rho_i \boldsymbol{u}_{EP,i} + \boldsymbol{J}_{diff,i}, \tag{5}$$

where $\boldsymbol{u}$ is the barycentric velocity, $\boldsymbol{u}_{EP,i}$ is the electrophoretic velocity, and $\boldsymbol{J}_{diff,i}$ is the diffusive flux. In the following sections, we define the models of the diffusive flux, the electrophoretic flux, and the barycentric flux.





### 2.1.2 Modeling of the diffusive flux


We use standard Fickian diffusion to model the diffusive flux,

$$\boldsymbol{J}_{\mathrm{diff},i} = -D_i M_i \nabla C_i \quad i = 1, 2, ..., N-1,$$ (6)

where $D$ is the diffusivity, $M$ is the molar mass, and $C$ is the molar concentration. The contribution of the diffusing solvent can be neglected, considering $\rho_i / \rho_N \ll 1$. Such an assumption leads to a particular choice of the $N$th component, which 85 contributes the most mass to the fluid. In our experiments of interest, the $N$th component is $\mathrm{H_2O(l)}$. Fickian diffusion utilizes the diagonal-diffusivity-matrix assumption, which is valid for many electrolytes (Dreyer et al., 2013). A diagonal diffusivity matrix means that short-range interactions between ions are negligible, which restricts our model to dilute solutions (Kontturi et al., 2008).

### 2.1.3 Modeling of the barycentric flux

The type of experiments of interest for this work is performed in a Hele-Shaw cell, for which fluid flow can be described by Darcy's law,

$$\boldsymbol{u} = -\frac{k}{\eta} \left( \nabla p - \rho \boldsymbol{g} \right),$$ (7)

where $k$ is the permeability, $\eta$ is the dynamic viscosity of the fluid, and $\boldsymbol{g}$ is the gravitational acceleration. When combined with the incompressibility assumption,

$$\nabla \cdot (\rho \boldsymbol{u}) = 0,$$ (8)

$\boldsymbol{u}$ and $p$ can be obtained. The scalar permeability of a Hele-Shaw cell, with a gap width of $w$, is given by

$$k = \frac{w^2}{12}.$$ (9)

### 2.1.4 Modeling of the electrophoretic flux

When the components in the fluids are electrically charged, for example, the ions $\mathrm{Na^+}$ or $\mathrm{Cl^-}$, we have to consider the effect 100 of Coulombic forces among the ions. Such Coulombic forces are typically modeled by incorporating the electric field $\boldsymbol{E}$ into the mass fluxes of the fluid components (Boudreau et al., 2004). The electrophoretic flux is described by

$$\rho_i \boldsymbol{u}_{\mathrm{EP},i} = \frac{z_i \rho_i D_i F}{RT} \boldsymbol{E},$$ (10)

where $z_i$ is the charge number of the fluid component, $F$ is the Faraday constant, $R$ is the ideal gas constant, and $T$ is the fluid temperature. We require the charge conservation equation to define the electric field in a multicomponent fluid. To obtain the 105 charge conservation equation of the ionic species, it is useful to use the flux defined in molar concentration. Using $\rho_i = M_i C_i$, we have the mass conservation equation (without considering the source/sink terms introduced by reactions)

$$\frac{\partial C_i}{\partial t} = -\nabla \cdot \left( C_i \boldsymbol{u} + C_i \boldsymbol{u}_{\mathrm{EP},i} + \boldsymbol{J}_{\mathrm{diff},i}/M_i \right).$$ (11)





Following the derivations of Kirby (2010), the charge density is defined as

$$\rho_{\mathrm{E}} = F \sum_{i=1}^{N-1} z_i C_i \,. \tag{12}$$

The charge conservation equation is the sum of Eq. (11) over all ionic species multiplied by the charge number and the Faraday constant,

$$\frac{\partial \rho_{\mathrm{E}}}{\partial t} = -F \, \nabla \cdot \left( \sum_{i=1}^{N-1} z_i C_i \boldsymbol{u} + z_i C_i \boldsymbol{u}_{\mathrm{EP},i} + z_i \boldsymbol{J}_{\mathrm{diff},i}/M_i \right) . \tag{13}$$

We assume the aqueous solution is locally electroneutral,

$$\sum_{i=1}^{N-1} z_i C_i = 0 \,, \tag{14}$$

and no charge accumulates at the continuum scale of interest,

$$\frac{\partial \rho_{\mathrm{E}}}{\partial t} = 0 \,. \tag{15}$$

The local electroneutrality condition is an approximation considering the Debye length (typically on the order of nanometers) vanishes at the length scale of the considered system (Dickinson et al., 2011). The electroneutrality assumption eliminates the barycentric flux term in the charge conservation equation. We consider Fickian diffusive flux, Eq. (6), so that the charge

conservation is given by

$$-\nabla \cdot \left( \sum_{i=1}^{N-1} \frac{D_i C_i (z_i F)^2}{RT} \boldsymbol{E} - \sum_{i=1}^{N-1} D_i z_i F \nabla C_i \right) = 0 \,. \tag{16}$$

The multiplier of the electric field, $\boldsymbol{E}$, is the electrical conductivity ($\mathrm{C}^2 \mathrm{m}^{-3} \mathrm{kg}^{-1} \mathrm{s}$),

$$\sigma = \sum_{i=1}^{N-1} \frac{D_i C_i (z_i F)^2}{RT} \,. \tag{17}$$

When the diffusivities of all species are equal, $D_i = D$, the diffusive flux in the charge conservation equation is

$$D \sum_{i=1}^{N-1} z_i F \nabla C_i = D \nabla \left( F \sum_{i=1}^{N-1} z_i C_i \right) = 0 \,. \tag{18}$$

Then, the charge conservation equation yields Laplace's equation of the electric potential, i.e., $\nabla \cdot (\sigma \nabla \phi) = 0$, where $\boldsymbol{E} = -\nabla \phi$. However, the diffusivity of the chemical species varies with molecular size and can differ by up to an order of magnitude. Such variations in molecular diffusivity lead to diffusive fluxes that produce an electric field. The model that combines mass conservation, Eq. (11), and charge conservation, Eq. (16), is known as the Poisson–Nernst–Planck (PNP) model.

The PNP model aims at resolving both the electric potential and the molar concentrations, subject to the boundary conditions of the electric potential. The mathematical properties of the PNP model, with no external flow (Filipek et al., 2017), coupled

 

with Darcy flow (Herz et al., 2012; Ignatova and Shu, 2022), and coupled with Navier–Stokes flow (Lee, 2021; Constantin et al., 2021, 2022) are still being explored. Rigorous upscaling and homogenization of the PNP–Stokes system has been performed (Ray et al., 2012a; Kovtunenko and Zubkova, 2021), and the upscaled formulations are examined by numerical modeling (Frank et al., 2011; Ray et al., 2012b). Numerical methods and solution techniques of the PNP equations are an ongoing research field (Flavell et al., 2014; Song et al., 2018; Shen and Xu, 2021; Liu and Maimaitiyiming, 2021; Yan et al., 2021; Liu et al., 2022; Zhang et al., 2022).

If there are no sources of the electric field (Tabrizinejadas et al., 2021), then the electric field can be represented by

$$\boldsymbol{E} = \frac{RT}{F} \frac{\sum_{j=1}^{N-1} D_j z_j \nabla C_j}{\sum_{k=1}^{N-1} (z_k)^2 D_k C_k}, \tag{19}$$

where we use $j$ and $k$ as summation indices. Hence, the molar electrophoretic flux of individual species can be formulated as

$$C_i \boldsymbol{u}_{\mathrm{EP},i} = \frac{z_i D_i C_i F}{RT} \boldsymbol{E} = z_i D_i C_i \frac{\sum_{j=1}^{N-1} z_j D_j \nabla C_j}{\sum_{k=1}^{N-1} (z_k)^2 D_k C_k}. \tag{20}$$

There are many physical interpretations of such a flux. For example, one is that the dissolved species is advecting due to the electric field. Another point of view can be obtained by combining Eq. (20) with the Fickian diffusion flux,

$$\boldsymbol{J}_{\mathrm{diff},i}/M_i + C_i \boldsymbol{u}_{\mathrm{EP},i} = -D_i \nabla C_i + \frac{z_i D_i C_i}{\sum_{k=1}^{N-1} (z_k)^2 D_k C_k} \sum_{j=1}^{N-1} z_j D_j \nabla C_j \tag{21}$$

$$= -\sum_{j=1}^{N-1} D_{ij} \nabla C_j, \tag{22}$$

which corresponds to the "natural generalization" of Fickian diffusion, denoted by Onsager (1945). Following Boudreau et al. (2004), the cross-coupling diffusivities are given by

$$D_{ij} = \delta_{ij} D_i - \frac{z_i z_j D_i D_j C_i}{\sum_{k=1}^{N-1} (z_k)^2 D_k C_k} = \delta_{ij} D_i - \frac{z_i D_i C_i \otimes z_j D_j}{\sum_{k=1}^{N-1} (z_k)^2 D_k C_k}, \tag{23}$$

where Einstein's summation convention is not applied. Hence, locally electroneutral diffusion is equivalent to nonlinear multicomponent diffusion with a diffusivity tensor being a rational function of molar concentrations of the charged species (Rubinstein, 1990). Similar forms of the cross-coupling diffusivities can also be found in Lichtner (1985).

Combining Eq. (11) and Eq. (21) yields

$$\frac{\partial C_i}{\partial t} = -\nabla \cdot \left( -D_i \nabla C_i + z_i D_i C_i \frac{\sum_{j=1}^{N-1} z_j D_j \nabla C_j}{\sum_{k=1}^{N-1} (z_k)^2 D_k C_k} + \boldsymbol{u} C_i \right), \tag{24}$$

which is the Nernst–Planck equation for single-phase-multicomponent systems. If all species have the same diffusivity, $D_i = D$, and the aqueous solution is electroneutral, Eq. (14). Then, the single-phase-multicomponent Nernst–Planck equation reduces to

$$\frac{\partial C_i}{\partial t} = -\nabla \cdot (-D \nabla C_i + \boldsymbol{u} C_i), \tag{25}$$





which we refer to as the single-diffusivity model. Both the Nernst–Planck model and the single diffusivity model satisfy the electroneutrality condition, Eq. (14), and the zero-charge accumulation assumptions, Eq. (15). However, of the two models,

only the Nernst–Planck model is capable of handling the physics of varying species diffusivities. To communicate the effectiveness of the Nernst–Planck equation in reactive transport scenarios, we compare the modeling results with the single diffusivity model, which is a common approach. We set the diffusivity of the single diffusivity model as $3 \times 10^{-3} \, \mathrm{mm^2 \, s^{-1}}$, which is within the common values for species diffusivities in water at $25 \, ^\circ\mathrm{C}$.

### 2.1.5 Chemical equilibrium using Gibbs-energy minimization

Another aspect of mass balance amongst the fluid components, which we now denote as the chemical species, is the reaction that changes the amount of the chemical species while conserving the mass balance of chemical elements. The reaction source/sink term in the mass balance of each fluid component, $Q_i$, is implicitly formulated as the Gibbs-energy minimization problem,

$$\min_n \ G = \sum_{i=1}^{N} \mathrm{n}_i \mu_i \quad \text{subject to} \begin{cases} \text{mass balance of chemical elements} \\ \mathrm{n} \geq 0 \end{cases}, \tag{26}$$

where $G$ is the Gibbs energy and $\mathrm{n}_i$ is the molar amount of the $i$th chemical species. The chemical potential is defined as

$$\mu_i = \mu_i^\circ + RT \log a_i, \tag{27}$$

where $\mu_i^\circ$ is the standard chemical potential of the $i$th species, and $a_i$ is the activity of the $i$th species. For the standard chemical potential, we employ the SUPCRT07 database (Johnson et al., 1992). To evaluate the activities, we use the Helgeson-Kirkham-Flowers (HKF) activity model (Helgeson and Kirkham, 1974a, b, 1976; Helgeson et al., 1981), which is based on the

extended Debye–Hückel activity model. For the activity model of $CO_2(\mathrm{aq})$, we use the Drummond model (Drummond, 1981). The combination of the SUPCRT database and the HKF activity model is utilized in many geochemical modeling packages, such as EQ3/6 (Wolery, 1992), CHILLER (Reed, 1998), ChemApp (Petersen and Hack, 2007), Perplex (Connolly, 2009), GEM-Selektor (Kulik et al., 2012), PHREEQC (Parkhurst and Appelo, 2013), Geochemist's Workbench (Bethke, 2007), and Reaktoro (Leal, 2022) (Miron et al., 2019). General reviews of the activity models can be found in Marini (2007); Hingerl

180  (2012).

We use Reaktoro to solve the chemical equilibrium problem. The numerical methods and implementations of the Gibbs-energy minimization problem are detailed in Leal et al. (2014, 2016, 2017). The equilibrium calculations are performed at every time step and for every discretized volume of the simulation domain, and calculations of similar chemical states can be cached for efficiency. Efficient lookup tables and machine learning approaches of chemical-equilibrium calculations are in

active research (Huang et al., 2018; De Lucia and Kühn, 2021; De Lucia et al., 2021; Savino et al., 2022; Laloy and Jacques, 2022), and significant speedups in solving reactive transport problems have been achieved (Kyas et al., 2022; Bordeaux-Rego et al., 2022). In particular, Reaktoro employs on-demand machine learning and physics-based interpolation, which is able to conserve mass and control interpolation accuracy (Leal et al., 2020). Surrogate models of the full reactive transport process,




using machine learning, have also been developed (Sprocati and Rolle, 2021). However, to avoid approximation errors, machine
learning approaches and surrogate models are not employed in this work.

## 2.2 Numerical schemes

This section elaborates on the numerical schemes we use for solving the fundamental equations of the reactive transport
processes.

### 2.2.1 The barycentric flux

We utilize the mixed finite element formulation to solve for fluid velocity and fluid pressure under the constraints of mass
balance, Eq. (8) and momentum balance, Eq. (7). To avoid spurious solutions, the finite element spaces have to satisfy the
Ladyzhenskaya-Babuška-Brezzi (LBB) condition. We refer the reader to other works for further discussions regarding the
LBB condition in the context of fluid flow problems (Donea and Huerta, 2003), geodynamics (Thieulot and Bangerth, 2022),
and elasticity and electromagnetic applications (Boffi et al., 2013). We select LBB-compatible elements, namely the zeroth-
order Raviart-Thomas (RT) element for the barycentric flux and the piecewise constant element for the pressure (Raviart and
Thomas, 1977). Another popular LBB-compatible element is the Brezzi-Douglas-Marini element (Brezzi et al., 1985). The
mixed finite element formulation can be written in matrix form as

$$
\begin{pmatrix} A & B^T \\ B & 0 \end{pmatrix} \begin{bmatrix} \boldsymbol{u} \\ p \end{bmatrix} = \begin{bmatrix} f \\ 0 \end{bmatrix}, \tag{28}
$$

which leads to a saddle point problem. Following Benzi et al. (2005)'s review of saddle point problems, one can apply Uzawa's
method (Uzawa, 1958) to solve such problems. Given an initial condition of $p^n$ at step $n$, Uzawa's method solves $\boldsymbol{u}$ and $p$
iteratively,

$$
\boldsymbol{u}^{n+1} = A^{-1}(f - B^T p^n), \tag{29}
$$

$$
p^{n+1} = p^n + \omega B \boldsymbol{u}^{n+1}, \tag{30}
$$

where $\omega > 0$ is a relaxation parameter. Normally, Uzawa's method converges slowly and requires many iterations, so that
improving Uzawa's method is an ongoing research effort (Bacuta, 2006). We use one of the improved methods, the augmented
Lagrangian Uzawa's method, introduced by Fortin and Glowinski (1983):

$$
\begin{pmatrix} A + rB^T B & B^T \\ B & 0 \end{pmatrix} \begin{bmatrix} \boldsymbol{u} \\ p \end{bmatrix} = \begin{bmatrix} f \\ 0 \end{bmatrix}, \tag{31}
$$

where $r > 0$ is a tuning parameter. We solve the matrix problem iteratively,

$$
\boldsymbol{u}^{n+1} = (A + rB^T B)^{-1}(f - B^T p^n), \tag{32}
$$

$$
p^{n+1} = p^n + \omega B \boldsymbol{u}^{n+1}. \tag{33}
$$




When $\omega = r$, we can choose sufficiently large $r$ to accelerate the convergence of Uzawa's method. However, the matrix $A + rB^TB$ becomes more ill-conditioned as $r$ increases. Hence, we use the MUMPS (Amestoy et al., 2001, 2019) direct solver to update the velocity, Eq. (32).

### 2.2.2 Transport of fluid components

In the previous section, we selected an LBB-compatible velocity and pressure pair. The basis function of pressure is a piecewise constant, which can be described as the average pressure of a certain cell volume in a given mesh. Therefore, it is straightforward to define the molar concentrations as piecewise constants. We use the finite volume method to discretize the transport of the fluid components, Eq. (24):

$$\int_\Omega q_i \frac{\partial C_i}{\partial t} d\Omega = -\int_\Gamma q_i \left( -D_i \nabla C_i + z_i D_i C_i \frac{\sum_{j=1}^{N-1} z_j D_j \nabla C_j}{\sum_{k=1}^{N-1} (z_k)^2 D_k C_k} + \boldsymbol{u} C_i \right) \cdot \boldsymbol{n} \, d\Gamma, \tag{34}$$

where the fluxes are yet to be defined. Considering no-flux boundary conditions of the fluid components, we use the two-point flux approximation scheme (TPFA) for the diffusive flux,

$$\int_\Gamma q_i D_i \nabla C_i \cdot \boldsymbol{n} \, d\Gamma = -\int_{\Gamma_{\text{int}}} [q_i] D_i \frac{[C_i]}{h} \, d\Gamma, \tag{35}$$

where $\Gamma_{\text{int}}$ denotes the interior boundaries, $[\bullet]$ is the jump operator, and $h$ is the distance between the cell centers. We assumed that the diffusivity of each species is constant. For diffusivities varying in space, Tournassat et al. (2020) discussed and com-

pared various averaging approaches of the diffusivities at the interfaces between cells. For the advective flux, we utilize the full upwinding method,

$$-\int_\Gamma q_i C_i \, \boldsymbol{u} \cdot \boldsymbol{n} \, d\Gamma = -\int_{\Gamma_{\text{int}}} [q_i] C_i^{\text{up}} \, \boldsymbol{u} \cdot \boldsymbol{n} \, d\Gamma, \tag{36}$$

where $C_i^{\text{up}}$ is the molar concentration in the upwind direction. For the Nernst–Planck term, we follow Tournassat et al. (2020)'s approach:

$$-\int_\Gamma q_i z_i D_i C_i \frac{\sum_{j=1}^{N-1} z_j D_j \nabla C_j}{\sum_{k=1}^{N-1} (z_k)^2 D_k C_k} \cdot \boldsymbol{n} \, d\Gamma = \int_{\Gamma_{\text{int}}} [q_i] z_i D_i \{C_i\} \frac{\sum_{j=1}^{N-1} z_j D_j [C_j]}{\sum_{k=1}^{N-1} (z_k)^2 D_k \{C_k\} h} \, d\Gamma, \tag{37}$$

where $\{\bullet\}$ is the averaging operator. We use the arithmetic average to represent the molar concentration at the cell interface, which is suitable when all charged components are mobile (Gimmi and Alt-Epping, 2018), or when there are no membrane effects or large concentration gradients (Tournassat et al., 2020).

With regard to the time-stepping schemes, we use an explicit scheme for the upwind advection term and the Crank–Nicolson

scheme for the diffusion and Nernst–Planck terms. To avoid negative concentration values, we perform a transformation of variables $C_i = \exp(c_i)$, which leads to the logarithmically transformed Nernst–Planck equations (Kirby, 2010). The transport equations are solved using PETSc (Balay et al., 2022).



### 2.2.3 Coupling flow, transport, and chemical equilibrium

Of many coupling approaches in reactive transport modeling (Steefel and MacQuarrie, 2018; Carrayrou et al., 2004; Abd and Abushaikha, 2021), we apply the sequential non-iterative approach (SNIA) to couple flow, transport, and chemical equilibrium. Figure 1 illustrates the simulation procedure, where the fluid flow, the species transport, and the equilbrium steps are solved consecutively.

Due to the simplicity and effectiveness in applying the SNIA coupling between the transport solvers and the chemical equilibrium codes, this coupling method is widely adopted in the following software: OpenGeoSys (Shao et al., 2009; Naumov et al., 2022), poreReact (coupling of OpenFOAM (Weller et al., 1998) and Reaktoro) (Oliveira et al., 2019), CSMP++GEM (Yapparova et al., 2019), FEniCS–Reaktoro (Damiani et al., 2020), Osures (Moortgat et al., 2020), FEniCS-based Hydro-Mechanical-Chemical solver (Kadeethum et al., 2021), PorousFlow (based on the MOOSE Framework (Permann et al., 2020)) (Wilkins et al., 2021), IC-FERST-REACT (Yekta et al., 2021), COMSOL and PHREEQC (Jyoti and Haese, 2021), GeoChem-Foam (coupling of OpenFOAM and PHREEQC) (Maes and Menke, 2021), coupling of Reaktoro and Firedrake (Rathgeber et al., 2016) (Kyas et al., 2022), and P3D-BRNS (Golparvar et al., 2022). For reviews of reactive transport codes and the underlying coupling approaches, we refer the reader to the publications by Gamazo et al. (2015); Damiani et al. (2020).

### 2.3 Selected experiments for model evaluation

We select three experiments, performed (by others) in a Hele-Shaw cell, to evaluate the effectiveness of the Nernst–Planck model in reactive transport situations. The diffusivities of the aqueous species are listed in Table 1. The density and dynamic viscosity of the relevant aqueous solutions are listed in Table 2. Our models assume constant dynamic viscosity, set as the average viscosity of the aqueous solutions. All experiments are performed in a lab environment, and we assume a constant temperature of 25 °C and a constant background pressure of 100 kPa. No-flow boundary conditions are prescribed on all sides of the simulation domain.

**Table 1.** Diffusivities of aqueous species in 25 °C water at infinite dilution

|  | $H^+$ | $OH^-$ | $Na^+$ | $Cl^-$ | $Li^+$ | $NO_3^-$ | $HCO_3^-$ | $CO_3^{2-}$ | $CO_2(aq)$ |
|---|---|---|---|---|---|---|---|---|---|
| $D\ (10^{-3}\,\mathrm{mm^2\,s^{-1}})$ | 9.311 | 5.273 | 1.334 | 2.032 | 1.029 | 1.902 | 1.101 | 0.804 | 2.045 |

The diffusivities of aqueous species other than $HCO_3^-$, $CO_3^{2-}$, and $CO_2(aq)$ are taken from Vanýsek (2021). The diffusivities of $HCO_3^-$, $CO_3^{2-}$, and $CO_2(aq)$ are calculated using Zeebe (2011).

### 2.3.1 Chemically driven convection of acid–base systems

Almarcha et al. (2010) experimentally showed that simple acid–base reactions, such as A + B → C, can lead to hydrodynamic instabilities due to differences in diffusivities and densities of the chemical species. For the first case in our numerical modeling study, we follow the experimental setup described in Almarcha et al. (2010). The acid (A) is 1 M HCl and the base (B) is



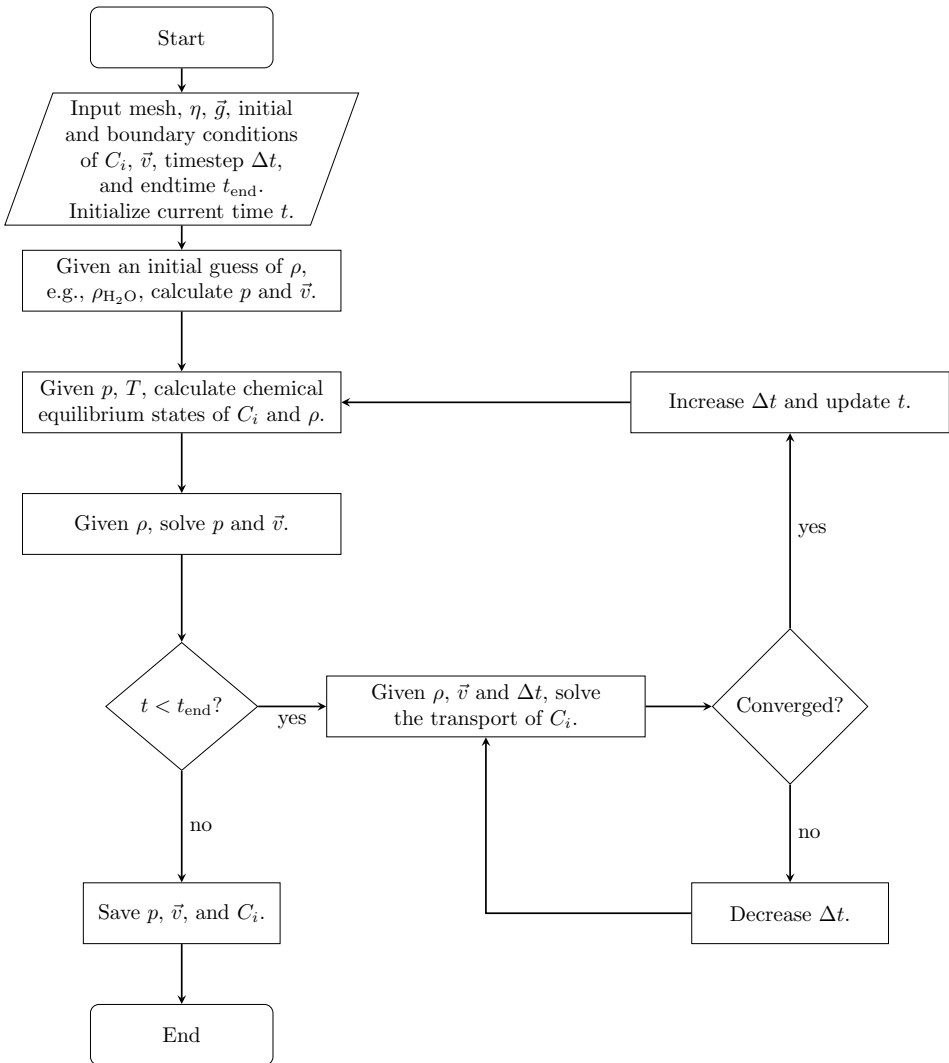

**Figure 1.** Flow chart of the simulation procedures that couple flow, transport, and chemical equilibrium, using the sequential non-iterative approach (SNIA).

1 M NaOH, where M is the molar concentration ($mol\,L^{-1}$). The experiment was performed by Almarcha et al. (2010) in a Hele-Shaw cell, which was $3.1\,cm$ wide, $5\,cm$ high, and had a $0.5\,mm$ gap width. To prevent Rayleigh-Taylor instability, the

270 less-dense acid was placed on top of the base. We consider the following species as the main fluid components in this case: $H^+$, $OH^-$, $Na^+$, $Cl^-$, and $H_2O(l)$. Please refer to the left panel of Figure 2 for the setup.

For the second modeling case, we choose a similar experiment, but with a strikingly different instability process. In their chemically driven convection experiments, Bratsun et al. (2017) reported a shock-wave-like structure of the acid–base interface.





**Table 2.** Dynamic viscosity and density of aqueous solutions and water at 25 °C

|            |                       | 1M HCl     | 1M NaOH    | 1.4M NaOH  | 1.5M HNO$_3$ | 0.01 M LiOH | H2O(l) |
| ---------- | --------------------- | ---------- | ---------- | ---------- | ------------ | ----------- | ------ |
| Literature | $\eta$ (mPa s)        | 0.959[a]   | 1.129[b]   | 1.232[b]   | 0.933[c]     | 0.893[b]    | 0.89   |
| Literature | $\rho$ (g/cm$^3$)     | 1.014[d]   | 1.039[b]   | 1.055[b]   | 1.046[c]     | 0.997[b]    | 0.997  |
| Reaktoro   | $\rho$ (g/cm$^3$)     | 1.012      | 1.037      | 1.053      | 1.044        | 0.997       | 0.997  |

Linearly interpolated using [a]Nishikata et al. (1981), [b]Sipos et al. (2000), [c]Zaytsev and Aseyev (1992), and [d]Åkerlöf and Teare (1938). The viscosity and density of water are taken from Lemmon and Harvey (2021).

In further studies, this phenomenon has subsequently been reproduced using many different acid–base pairs (Mizev et al., 2021; Bratsun et al., 2021). For our numerical comparison study, we use the acid-base pair from Bratsun et al. (2021), i.e., HNO$_3$–NaOH. The acid consists of 1.5 M HNO$_3$ and is placed on top of 1.4 M NaOH. The Hele-Shaw cell of this experiment has a width of 2.5 cm, a height of 9.0 cm, and a gap width of 1.2 mm. We consider the following species as the main fluid components in this case: H$^+$, OH$^-$, Na$^+$, NO$_3^-$, and H$_2$O(l). The setup is shown in Figure 2. We perform the simulation in a 2D geometry with the same width, height, and species initial conditions as in the experiment, and compare the simulation results with the time series of interferometry snapshots taken during the experiments.

### 2.3.2 Convective dissolution of CO$_2$ in reactive alkaline solutions

Thomas et al. (2016) performed experiments of convective dissolution of CO$_2$ in reactive alkaline solutions. The experiment setup consisted of a Hele-Shaw cell, saturated with alkaline solutions, and gaseous CO$_2$ flows over the top of the Hele-Shaw cell. Eventually, gaseous CO$_2$ dissolves into the alkaline solution, leading to density changes in the solution and, as a result, the formation of convection cells. Furthermore, Thomas et al. (2018) studied the convective dissolution of CO$_2$ in various molar concentrations of NaCl solutions using the aforementioned experimental setup. Likewise, Mahmoodpour et al. (2019) studied the effect of various salt solutions on the onset of convective instabilities by conducting experiments within porous media. Similar experimental studies in a sand box have been performed using dense miscible fluids (Neufeld et al., 2010; Agartan et al., 2015; Guo et al., 2021; Wang et al., 2021; Tsinober et al., 2022). Although such experiments do capture the effects of density-driven flow, electromigration effects are not involved. Other experimental studies of convective dissolution of gaseous CO$_2$ in aqueous solutions have been conducted in pressure-volume-temperature (PVT) cells (Yang and Gu, 2006), Hele-Shaw cells (Kneafsey and Pruess, 2010; Class et al., 2020; Zhang et al., 2020; Jiang et al., 2020; Teng et al., 2021), and 3D granular porous media (Mahmoodpour et al., 2020; Brouzet et al., 2022). Amarasinghe et al. (2020) reviewed CO$_2$ dissolution experiments and performed experiments of convective dissolution of CO$_2$ in a Hele-Shaw cell and porous media under supercritical CO$_2$ conditions.

Numerical studies of CO$_2$ convective dissolution have been performed considering the reactive transport of: a single component (Azin et al., 2013), a single component with mineral reactions (Babaei and Islam, 2018; Shafabakhsh et al., 2021), a single component in two fluid phases with a resolved gas–fluid boundary (Martinez and Hesse, 2016), multiple components





with mineral reactions (Zhang et al., 2011), and multiple components in two fluid phases with mineral reactions (Audigane
et al., 2007; Soltanian et al., 2019; Sin and Corvisier, 2019). However, the Nernst–Planck model was not invoked in any of
these works.

Using Lattice-Boltzmann methods with consideration of electrostatic forces, Fu et al. (2020, 2021) investigated how ionic
species of various salt solutions with varying concentrations affect the onset time and mass transfer rate of $CO_2$ convective
dissolution. So far, these are the only works we found that account for electromigration effects when modeling the convective
dissolution of $CO_2$.

In the third case of our numerical study, we model the experiment of Thomas et al. (2016) and investigate the reactive
transport of $CO_2$ in a Hele-Shaw cell including the electromigration of species using the Nernst–Planck model. Instead of
comparing our results to the experiments at the full scale ($165\,\mathrm{mm}$ by $210\,\mathrm{mm}$) (Thomas et al., 2016), a simulation domain
with a width of $100\,\mathrm{mm}$, a height of $40\,\mathrm{mm}$, and a gap width of $0.5\,\mathrm{mm}$ is utilized, see the right panel of Figure 2. The smaller
simulation domain was chosen, as the relevant processes remained within this smaller domain. On the top boundary of the
simulation domain, we assume that the gaseous $CO_2$ and the alkaline brine are always in equilibrium. The alkaline brine is
0.01 M LiOH. We consider the following species as the main fluid components in the case of convective dissolution of $CO_2$:
$H^+$, $OH^-$, $Li^+$, $CO_2(aq)$, $HCO_3^-$, $CO_3^{2-}$, and $H_2O(l)$. In the results section, we present the development of the fingers and
the movement of the "shock wave" induced by density, reaction, and electromigration effects.

## 3  Results

This section compares the simulations and the experimental results of the chemically driven convection problems and the
convective dissolution of $CO_2$ into a reactive alkaline solution. All simulations are performed using the Nernst–Planck and the
single-diffusivity models.

### 3.1  Comparing the chemically driven convection experiments with the single-diffusivity model and the
Nernst–Planck model

We present the results of the chemically driven convection between 1 M HCl and 1 M NaOH solutions. Figure 3 compares the
digital interferometry images obtained from experiments with the simulated fluid densities. The interferometry images have
a size of $13\,\mathrm{mm}$ by $23.075\,\mathrm{mm}$, from which we select a sub-region of the same size from the simulations. This region is
indicated by the red box in the right panel of Figure 3. Note that the interferometry images show the density contrast of the
fluids (induced by changes in refractive index), not the density itself. Hence, the simulated density plots cannot be compared
with the interferometry images pixel by pixel. Nonetheless, in the left panel of Figure 3, only the simulations with the Nernst–
Planck model replicate the experimentally observed fingering effect with a similar number of fingers. At the simulation time of
60 seconds, a low-density layer (shown in yellow) emerges in the Nernst–Planck model, an effect which cannot be observed in
the single-diffusivity model. At 115 seconds, the low-density layer develops instabilities, causing density-driven flow. However,
the time it takes for fingering to commence is longer in the simulations than in the experiments. This delay can be attributed





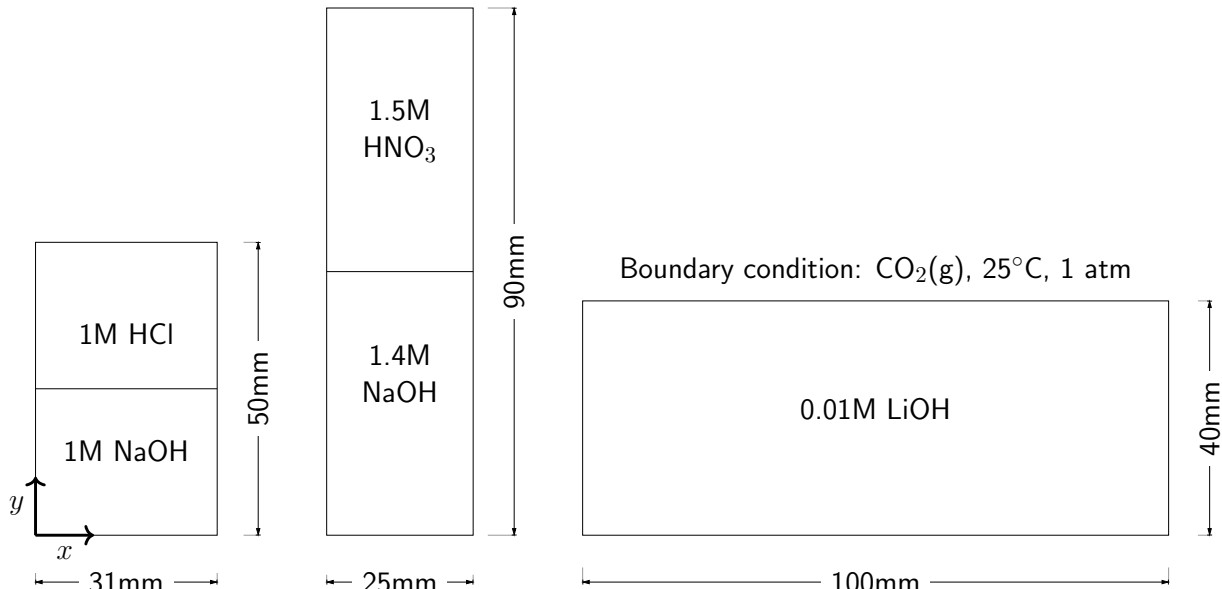

**Figure 2.** The selected experimental setups: The left panel shows the chemically driven convection experiments with 1 M HCl–1 M NaOH (Almarcha et al., 2010). The middle panel shows the chemically driven convection setup with 1.5 M HNO₃–1.4 M NaOH (Mizev et al., 2021). The right panel shows a smaller setup of CO₂ dissolution with 0.01 M LiOH solutions, compared to the original setup (165 mm by 210 mm), designed by Thomas et al. (2016). From left to right, the gap widths of the Hele-Shaw cells are 0.5 mm, 1.2 mm, and 0.5 mm. All Hele-Shaw cells are closed, so that no fluid leaves the domain.

to the finite representation by the numerical scheme of the diffusion and electromigration fluxes at the initially sharp interface between the acid and the base, which could be improved by adaptive mesh refinement or a higher mesh resolution.

Figure 4 compares the images from the physical experiment with our numerical simulations that utilize the Nernst–Planck model with a lag time of 40 seconds. We further present the filled contours of the $Na^+$ molar concentration and the molar concentration of $Na^+$ and $Cl^-$ combined. $Na^+$ is transported upwards, closely following the low-density fingers. The rightmost plots of Figure 4 show that the low-density region is caused by lower concentrations of both $Na^+$ and $Cl^-$. The dark red region indicates lower salinity and corresponds to the yellow-turquoise, low-density region. More interestingly, a stable region of higher salinity (shown in blue) forms close to the initial contact line of the acid and the base.

Figure 5 shows a comparison of the experiments of 1.5 M HNO₃ and 1.4 M NaOH solutions and the simulations. Both the Nernst–Planck model and the single-diffusivity model reproduce the shock-wave-like structures seen in the experimental interferometry images, i.e., the rapid downward movement of the acid–base interface. In the interferometry images at the 3 seconds snapshot, fingers appear, which the simulations cannot reproduce at the 3 seconds simulation time. As discussed before, during the HCl–NaOH experiments, the mismatch is due to the poor approximation of diffusive fluxes at the initially



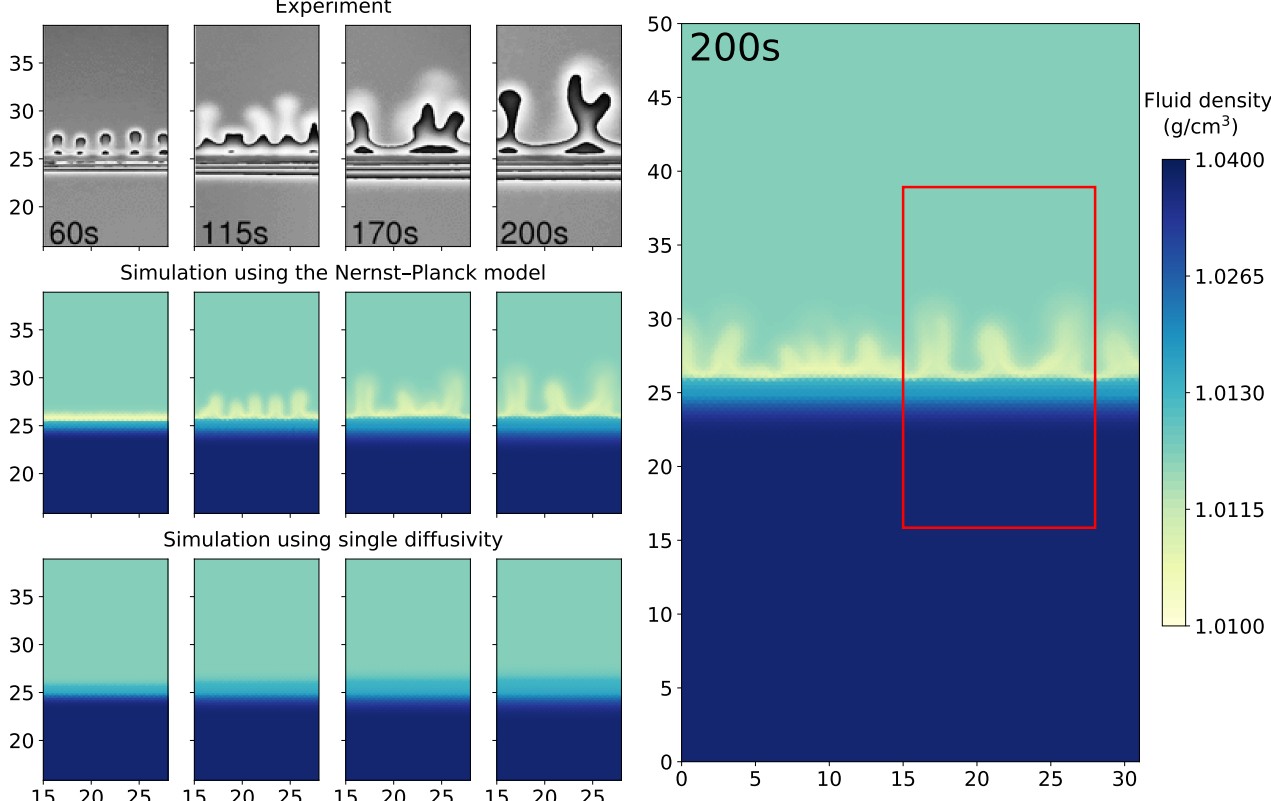

**Figure 3.** Comparing the simulations using the Nernst–Planck model and those using the single-diffusivity model with the experiments of chemically driven convection in HCl–NaOH solutions. The right panel shows the full simulation domain (31 mm by 50 mm), and the red box indicates the comparison window. On the left panel, the time increases from left to right, and the simulation time corresponds to the time shown in the experiment snapshots. The experimental figures are based on digital interferometry, where we adapted the figures from Almarcha et al. (2010).

sharp interface. For the later snapshots, 150 and 700 seconds, we observe the sparsening of the dark lines in the interferometry

images. This indicates that the fluid densities become more homogeneous, and the simulations replicate such mixing physics.

In Figure 6, we show the simulations of the molar concentrations of $Na^+$ and the streamlines of the fluid velocity field at 25, 50, 75, 100, and 125 seconds. One can clearly see that the $Na^+$ plumes follow the streamlines. The maximum velocity in the $y$ direction of the Nernst–Planck simulations and the single-diffusivity simulations during the five presented snapshots are $4.086\,\mathrm{mm/s}$ and $4.184\,\mathrm{mm/s}$, respectively. Both models replicate the experiments, and no particular difference can be

observed. In such experimental settings, where density-driven convection is dominant, a distinctive description of the diffusive fluxes is less crucial.





**Figure 4.** Comparing the simulation results using the Nernst–Planck model with the experimental results of chemically driven convection in HCl–NaOH solutions. The leftmost column shows the experimental figures, adapted from Almarcha et al. (2010). The filled contour plots present the simulation results, showing the fluid density, molar concentration of $Na^+$, and the sum of the molar concentrations of $Na^+$ and $Cl^-$ with a lag time of 40 seconds.

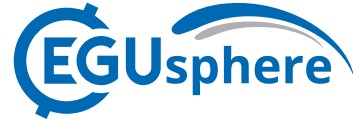



**Figure 5.** Comparing the simulations using the Nernst–Planck model and those using the single-diffusivity model with the experiments of chemically driven convection in $HNO_3$–$NaOH$ solutions. The right panel shows the full simulation domain ($25\,mm$ by $90\,mm$), and the red box indicates the comparison window. In the left panel, the time increases from left to right, where the snapshots correspond to 3, 150, and 700 seconds. The figures from the experiments are based on digital interferometry, and were adapted from Mizev et al. (2021).





**Figure 6.** Comparing the simulations using the Nernst–Planck model and those using the single-diffusivity model in modeling chemically driven convection of HNO₃–NaOH solutions. The top and the bottom panel show the simulations employing the Nernst–Planck model and the simulations using one single diffusivity, respectively. The time increases from left to right, and the snapshots correspond to 25, 50, 75, 100, and 125 seconds.



## 3.2 Simulation of convective dissolution of $CO_2$ in reactive alkaline solutions

Figure 7 compares the schlieren images of convective dissolution of $CO_2$ in 0.01 M LiOH solution from the experiments with the density plots using the Nernst–Planck and the single-diffusivity models. The onset time of the Nernst–Planck model and the single-diffusivity model are approximately 600 and 1400 seconds, respectively. Therefore, the densities of the two simulation results are shown for the onset time plus the experimental snapshot, 780 seconds. At these time steps, the convective fingers in both models reach similar depths of $\sim 17\,\mathrm{mm}$, as shown in the schlieren image from the experiment. The simulation result of the Nernst–Planck model is comparable to the experiment of $CO_2$ dissolving into the LiOH brine and shows a broader range of finger sizes compared to those of the single-diffusivity model. In contrast, the convective fingers in the single-diffusivity model resemble the fingers seen in the schlieren image, where $CO_2$ dissolves in $H_2O$, where the electromigration effects are weaker, compared to the LiOH solution.

Figure 8 shows the density plots of the simulation results using the Nernst–Planck model and the single-diffusivity model. In the top panel, we show the onset of the convective instability. The single-diffusivity model develops instabilities roughly 800 seconds later than the Nernst–Planck model. The second panel from the top shows the convective fingers after their onset; the Nernst–Planck model yields twice the number of fingers, compared to the single-diffusivity model. In these two panels, we observe that the Nernst–Planck model generates a low-density layer, colored in yellow, at the dissolution front, which causes a shorter onset time and more fingers due to a larger density contrast. In the two bottom panels, we show the evolution of the simulations at later times. There are differences in the fingering pattern, such as more secondary convective fingers being generated by the Nernst–Planck model. Moreover, the low-density layer in the Nernst–Planck model persists even at later time.

To further quantify the simulation results, Figure 9 visualizes the pH plots as well as the molar concentrations of $Li^+$, $CO_2(aq)$, and $CO_3^{2-}$. We use a diverging color palette to present the acid regions (orange) and the base regions (purple). The molar concentrations of $CO_2(aq)$ and $CO_3^{2-}$ follow the pH color palette, where most $CO_2(aq)$ exists in the acidic region and most $CO_3^{2-}$ exists in the basic region. Such changes in the $CO_2$ states lead to changes in the charge states $z = 0, 1, 2$ for $CO_2(aq)$, $HCO_3^-$, and $CO_3^{2-}$, respectively. In the $Li^+$ plot of the Nernst–Planck simulations, we observe up to 30% differences in the molar concentration compared to the initial condition ($0.01\,\mathrm{mol/L}$). The red regions are where $Li^+$ accumulates, and we can clearly see the interaction between $Li^+$ and $CO_3^{2-}$, whereby $Li^+$ migrates towards the double-charged $CO_3^{2-}$. Meanwhile, the single-diffusivity simulation shows almost no contrast in the filled contours, other than due to numerical errors. The difference between the maximum and minimum molar concentrations of $Li^+$ is $\sim 10^{-8}\,\mathrm{mol\,L^{-1}}$. Using the Nernst–Planck model, we confirm the influence of the inert (spectator) ions in the reactive-transport situation discussed in Thomas et al. (2016).

## 4 Discussion

In this section, we discuss whether the Nernst–Planck model is valid, and its differences are compared to the commonly used single-diffusivity model.

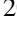


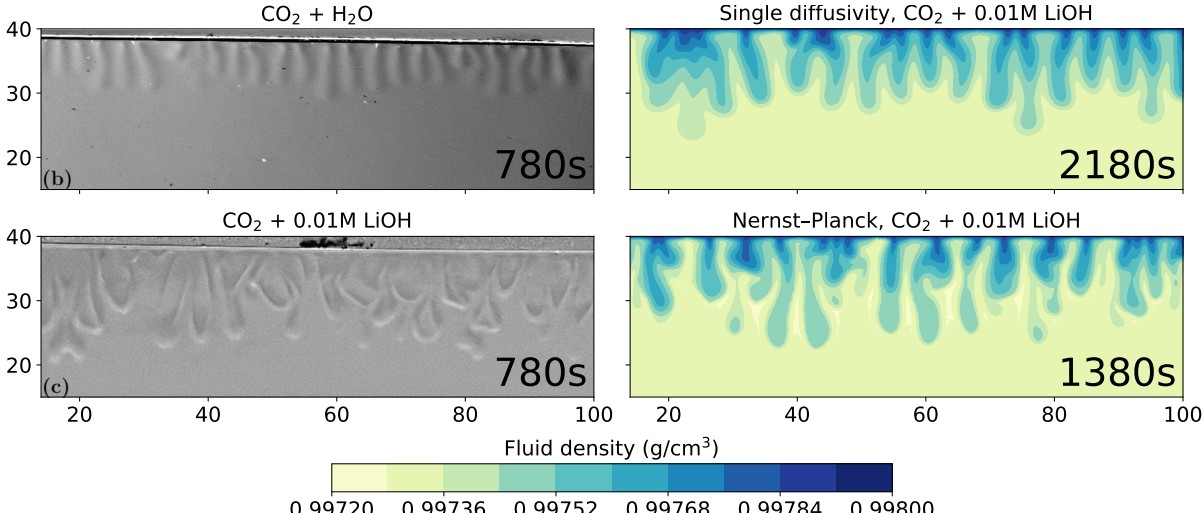

**Figure 7.** Comparing the simulations using the Nernst–Planck model and those using the single-diffusivity model with the experiments of convective dissolution of $CO_2$ in 0.01 M LiOH solution. We plot the experimental figure of $CO_2$ dissolving in pure water in the top-left to demonstrate the differences in fingering patterns, compared with $CO_2$ dissolving in LiOH solution. The filled contours show the fluid density. The experimental figures are adapted from Thomas et al. (2016).

### 4.1 The validity of the Nernst–Planck model under reactive flow conditions

Comparing to two chemically driven convection experiments conducted by Almarcha et al. (2010) and Mizev et al. (2021),
we show that the Nernst–Planck numerical model is able to reproduce the convective fingering observed in the experiments, especially in the 1 M HCl–1 M NaOH case. The mechanism that leads to convection is known as the reactive diffusive layer convection (DLC) instability, which arises when a less-dense solution of a fast-diffusing component overlies a denser solution of a slow-diffusing component (Lemaigre et al., 2013). In our Nernst–Planck model, the fast-diffusing component is $H^+$. Additionally, the convective plumes of the reactive DLC instability are only observed above the initial contact line between
the acid and the base (Almarcha et al., 2010; Lemaigre et al., 2013). This is the asymmetric feature of the reactive DLC instability, which can also be observed in the Nernst–Planck simulations (Figure 3). Directly quoting Lemaigre et al. (2013): "The asymmetry of the DLC pattern is related to the chemical reaction which consumes the acid before it can accumulate in the lower layer and replaces it by a third species of intermediate contribution to density". This third species is salt, a product of the acid–base neutralization reaction. Using the Nernst–Planck model, we not only replicate this stabilizing lower layer, but
also find that the accumulation of salt can exceed the salt concentration at chemical equilibrium (in this case, approximately $0.5 \, \text{mol} \, \text{L}^{-1}$), as shown in Figure 4.

In the 1.5 M $HNO_3$–1.4 M NaOH case, both the Nernst–Planck model and the single-diffusivity model reproduce the shock-wave-like structure observed in the experiments performed by Mizev et al. (2021). The instability in this case is convection-controlled (CC) (Bratsun et al., 2017). The underlying mechanism of this CC instability has previously been attributed to





**Figure 8.** Comparing the simulations, using the Nernst–Planck model and those using the single-diffusivity model, when numerically modeling convective dissolution of $CO_2$ in 0.01 M LiOH solution. The filled contours show the fluid density. Time increases from top to bottom and is indicated in the plots.




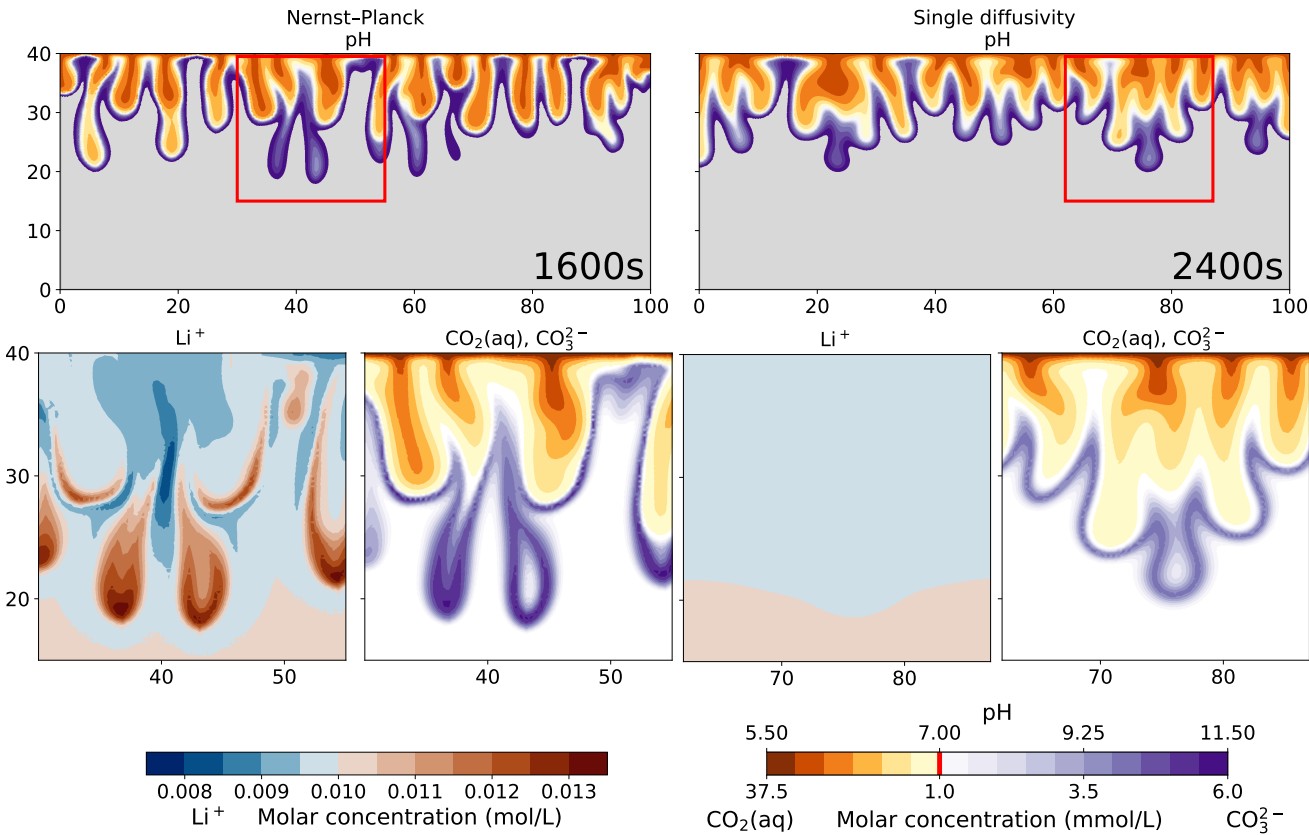

**Figure 9.** Comparing the simulations, using the Nernst–Planck model and those using the single-diffusivity model, when numerically modeling convective dissolution of $CO_2$ in $0.01\,M$ LiOH solution. The top panel shows the pH values. The orange palette represents acidic regions (pH$< 7$) and the purple palette represents basic regions (pH$> 7$). The gray region shows the unperturbed part of the LiOH solution (pH$> 11.5$). In the bottom panel, we show the zoomed-in view of the red boxes in the top panel. We plot the molar concentrations of $Li^+$, $CO_2(aq)$, and $CO_3^{2-}$. Notice that the molar concentrations of $CO_2(aq)$ and $CO_3^{2-}$ share the same color bar with the pH values. The red vertical line in the color bar indicates the minimum molar concentrations of $CO_2(aq)$ and $CO_3^{2-}$ shown in the plots.





unequal and concentration-dependent diffusivities (Bratsun et al., 2017, 2021, 2022). However, using the single-diffusivity model, our simulations can reproduce the experimental observations. This indicates that modeling CC instabilities is less sensitive to the specific diffusivity model used, as the CC instability is mostly caused by the density-driven convective flow of the reactants. Nevertheless, there are no drawbacks (other than being computationally more intensive) in employing the Nernst–Planck equations to model the CC instability cases.

In the case of $CO_2$ dissolving in 0.01 M LiOH solution, we observe a much later convective onset time in the numerical simulations than in the experiment by Thomas et al. (2016). This discrepancy is much larger than in the 1 M HCl–1 M NaOH case. It cannot only be explained by the ineffectiveness of the diffusive-flux approximations at the sharp interface of dissolving $CO_2(aq)$. Convective instability can be introduced in a variety of ways in laboratory experiments. For example, interfacial instabilities between the gas and the liquid can occur due to the gas flowing over the interface. Furthermore, if the injected $CO_2$

is not saturated with water, evaporation of water can induce gravitational instability (Bringedal et al., 2022). These localized physical processes at the gas-liquid boundary, which are not included in our numerical models, could be partly responsible for the later onset times we observe in our simulations.

    Although the comparison between the experiment and the simulations of the two models in Figure 7 can be ambiguous, i.e., the Nernst–Planck model is not *the* model that describes the schlieren images, we emphasize that the difference between

the experiments with $Li^+$ and those without it is apparent in the figure. When comparing the Nernst–Planck and the single-diffusivitiy models, the former exhibits an earlier fingering onset time. This is caused by electromigration effects, which generate the low-density layer shown in Figure 8. The electromigration of $Li^+$, an inert ion in this case, and subsequent effects on the overall transport processes shown in Figure 9, is clearly a motivation for more studies on the electromigration effects of inert (spectator) ions.

To summarize, we use the experiments conducted by Almarcha et al. (2010); Thomas et al. (2016); Mizev et al. (2021) and our numerical simulation results to show that the Nernst–Planck model is valid in the considered reactive transport scenarios. Particularly in the 1 M HCl–1 M NaOH experiments, where the convective fingers are controlled by diffusion. In the convection-controlled regime, electromigration effects are less crucial, and both the Nernst–Planck model and the single-diffusivity model replicate the 1.5 M $HNO_3$–1.4 M NaOH experiments well. In the case of $CO_2$ dissolving in an alkaline

solution, the use of the Nernst–Planck model uncovers intricate details on how inert ionic species can affect the overall reactive transport processes.

## 4.2   Electromigration and the relevance of inert ionic species

Using the simulations of chemically driven convection in 1 M HCl–1 M NaOH solutions, we elaborate the main differences between the Nernst–Planck model and the single-diffusivity model. The top panel of Figure 10 shows the molar concentra-

tions of salt, plotted against elevation in the Hele-Shaw cell. As expected, the single-diffusivity model shows the concentration profile of two diffusing Heaviside step functions of $Na^+$ and $Cl^-$. We plot the sum of $Na^+$ and $Cl^-$ molar concentrations to demonstrate that in single-diffusivity models, $Na^+$ and $Cl^-$ exhibit an ideal behavior, known as equimolar counterdiffusion. The Nernst–Planck model shows unexpected results, such as the accumulation of $Na^+$ and $Cl^-$ close to the acid–base bound-





ary. The accumulation can be attributed to a larger diffusivity of the acid components, $H^+$ ($9.311 \times 10^{-3}\,\mathrm{mm^2s^{-1}}$) and $Cl^-$

($2.032 \times 10^{-3}\,\mathrm{mm^2s^{-1}}$), compared to the base components, $OH^-$ ($5.273 \times 10^{-3}\,\mathrm{mm^2s^{-1}}$) and $Na^+$ ($1.334 \times 10^{-3}\,\mathrm{mm^2s^{-1}}$),
which leads to more $Cl^-$ migrating into the base. One might get the impression that this observation can be modeled by employing multiple diffusivities in the single-diffusivity model. However, we emphasize that the Nernst–Planck model has to
be utilized to maintain electroneutrality at the continuum scale. Closely inspecting the fluid-density profile, the deficit in salt
concentration leads to a lower density than the acid, which causes the reactive DLC instability. The single-diffusivity model is

unable to depict the accumulation or deficit in the salt concentrations.

Moreover, the only reaction that occurs in this experiment is the neutralization reaction

$$H^+ + OH^- \rightleftharpoons H_2O(l). \tag{38}$$

No reaction occurs for $Na^+$ and $Cl^-$. Hence, they are inert (spectator) species. We elaborate on the transport of $H^+$, given by

$$\frac{\partial C_{H^+}}{\partial t} = -\nabla \cdot \left( -D_{H^+H^+} \nabla C_{H^+} - D_{H^+OH^-} \nabla C_{OH^-} - D_{H^+Na^+} \nabla C_{Na^+} - D_{H^+Cl^-} \nabla C_{Cl^-} + \boldsymbol{u} C_{H^+} \right), \tag{39}$$

where the generalized-diffusion interpretation of the Nernst–Planck equation is applied, Eq. (22), and $D_{ij}$, are defined by
Eq. (23). In the Nernst–Planck model, the inert species contribute to the transport processes in the form of the cross-coupling
diffusivities, $D_{H^+Na^+}$, $D_{H^+Cl^-}$, and the concentration gradients, $\nabla C_{Na^+}$ and $\nabla C_{Cl^-}$. It has also been shown that the cross-coupling diffusivities (without reactions) can lead to hydrodynamic instabilities (Budroni, 2015). Such tight coupling between
ionic species does not exist in the single-diffusivity model. Hence, the Nernst–Planck model is more expressive and capable of

modeling the intricacies of reactive transport.

To further demonstrate the electromigration effects of the inert species, Figure 11 shows the species concentration profile
of simulations of $CO_2$ dissolution in 0.01 M LiOH solution. We plot the 600-seconds snapshot, which is the onset time of the
Nernst–Planck model (see Figure 8). Figure 11 focuses on the concentration profile close to the top boundary, which is the
elevation at 40 mm. When $CO_2$ dissolves into the LiOH solution, the following chemical reactions occur

$$CO_2(aq) + H_2O(l) \rightleftharpoons H_2CO_3(aq) \tag{40}$$

$$H_2CO_3(aq) \rightleftharpoons H^+ + HCO_3^- \tag{41}$$

$$H^+ + OH^- \rightleftharpoons H_2O(l), \tag{42}$$

where $H_2CO_3(aq)$ is a reaction intermediate that we have not considered in our numerical models. These reactions represent
the entire process of $CO_2(aq)$ acidifying aqueous solutions. If we add Eqs. (40)–(42), this yields a simplified reaction,

$$CO_2(aq) + OH^- \rightleftharpoons HCO_3^-. \tag{43}$$

In Figure 11, at elevations above 38 mm, excessive amounts of $CO_2(aq)$ react with $OH^-$, producing $HCO_3^-$. However, due to
the electroneutrality constraints, the concentration of $HCO_3^-$ is limited by the concentrations of the cations, particularly $Li^+$.
When $CO_2(aq)$ migrates deeper into the more basic regions, i.e., elevations between 36–38 mm, another reaction happens,

$$HCO_3^- \rightleftharpoons H^+ + CO_3^{2-}. \tag{44}$$


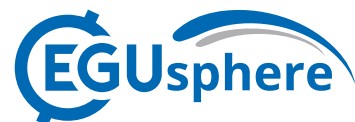

In this basic region, $HCO_3^-$ donates $H^+$ and forms $CO_3^{2-}$. We can still observe $Li^+$ constraining the amount of $CO_3^-$ owing to electroneutrality. Another cation that can contribute to electroneutrality is $H^+$. However, in this basic region, the amount of $H^+$ is negligible. Although $Li^+$ is inert and does not take part in the reactions, Eqs. (40)–(44), it is still relevant in the overall reactive transport process. The transport of $Li^+$ in the top-left panel of Figure 11 is evidence of its involvement in the reactive transport process.

Focusing on the density plots in Figure 11, not only does the Nernst–Planck model produce lower densities than the single-diffusivity model, the density gradient between $37$–$38\,mm$ is also higher in the Nernst–Planck model. This can explain the earlier onset time of the Nernst–Planck model, compared to the onset time of the single-diffusivity model. We end the discussion with Figure 12, which presents $Li^+$ migrating as $CO_2$ is dissolving and reacting in the solution. The tight coupling between the equilibrium reactions, electromigration effects, and density-driven flow elicits further investigation, both numerically and experimentally.

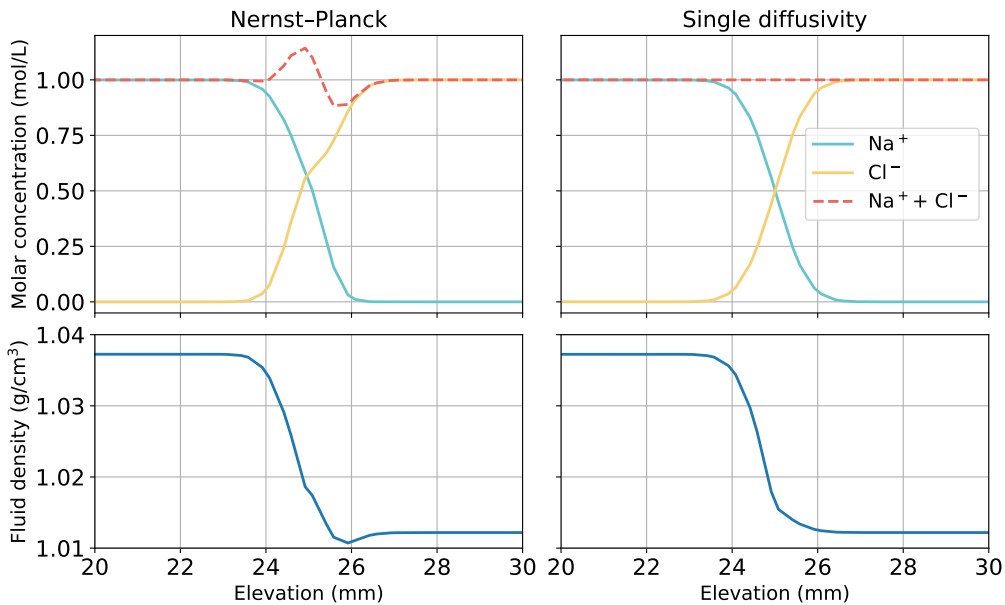

**Figure 10.** Comparing the simulations using the Nernst–Planck model and those using the single-diffusivity model in modeling chemically driven convection in $1\,M$ HCl–$1\,M$ NaOH solutions. The snapshot is given at $t = 60$ seconds, when fingers have not yet appeared (see Figure 3). We plot the molar concentrations of salt and the fluid density against elevation in the Hele-Shaw cell, focusing on the contact region between the acid and the base, $25\,mm$.


## 5 Conclusions

By comparing the single-diffusivity model and the Nernst–Planck model with reaction-driven flow experiments, we have demonstrated the importance and necessity under certain conditions of modeling the electromigration effects between charged



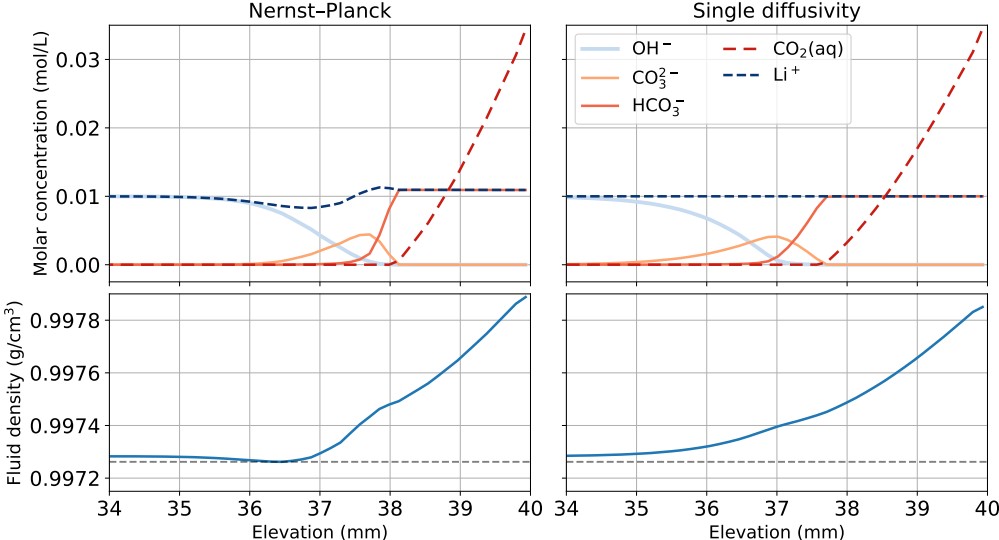

**Figure 11.** Comparing the simulations using the Nernst–Planck model and those using the single-diffusivity model in modeling convective dissolution of $CO_2$ in 0.01 M LiOH solution. The snapshot is given at $t = 600$ seconds, when fingers have not yet appeared (see Figure 8). We plot the molar concentrations of $Li^+$, $OH^-$, $CO_3^{2-}$, $HCO_3^-$, and $CO_2(aq)$ and the fluid density against elevation in the Hele-Shaw cell, focusing on the regions close to the top boundary of the Hele-Shaw cell. The gray-horizontal-dashed lines in the bottom panel show the minimum density observed in the Nernst–Planck model at $t = 600$ seconds.

species in aqueous environments using the Nernst–Planck model. Our results of simulating the reaction-driven flow exper-
iments and convective dissolution of $CO_2$ show that non-reacting species ($Na^+$, $Cl^-$, $Li^+$) influence the overall reactive
transport process via electromigration effects, which couple the charged species. Such electromigration effects cannot be mod-
eled using the single-diffusivity model, and further studies on quantifying the valid regimes of the single-diffusivity model are
recommended. We conclude that:

– By comparison of our numerical modeling results to previously published flow experiments, we have shown that the
Nernst–Planck model is valid for modeling reactive transport processes. The processes in the experiments considered
here are characterised by an intricate interplay between diffusion, reaction, electromigration, and density-driven convec-
tion.

– Compared to the often-used single-diffusivity model, the Nernst–Planck model enables the numerical modeling of the
electromigration of ionic species, introduced by differing species diffusivities, resulting in more and improved physical
insights.

– These reaction-driven flow experiments from literature can be further utilized for benchmarking reactive transport codes.





**Figure 12.** Simulation results using the Nernst–Planck model, showing the molar concentration of $Li^+$ during the convective dissolution of $CO_2$ in $0.01\,M$ LiOH solution.

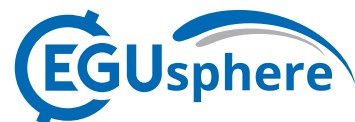

*Code and data availability.* The RetroPy code is available on GitHub (https://github.com/pwhuang/RetroPy), and version 1.0 is archived on Zenodo (https://doi.org/10.5281/zenodo.7371384). The RetroPy code is published under the GNU Lesser General Public License (LGPL). The codes that produced the simulations are located in the example folder of RetroPy, where the chemical_convection folder contains the chemically-driven convection examples, and the CO2_convection folder contains the $CO_2$ convective dissolution example. The data and scripts that recreate the figures are available on Zenodo (https://doi.org/10.5281/zenodo.7362225).

*Video supplement.* We provide the simulation results of Figure 4, 6, and 12 in the mp4 format. They are located in https://doi.org/10.3929/ethz-b-000579224.

*Author contributions.* Po-Wei Huang: Conceptualization, Methodology, Software, Formal analysis, Writing - Original Draft, Writing - Review & Editing, Visualization
Bernd Flemisch: Writing - Review & Editing
Chao-Zhong Qin: Writing - Review & Editing
Martin O. Saar: Supervision, Writing - Review & Editing
Anozie Ebigbo: Conceptualization, Writing - Review & Editing, Project administration, Funding acquisition

*Competing interests.* The authors declare that there is no known conflict of interest.

*Acknowledgements.* This work was supported by the Swiss National Science Foundation project entitled "Analysing spatial scaling effects in mineral reaction rates in porous media with a hybrid numerical model." We also thank the Werner Siemens-Stiftung (Werner Siemens Foundation) for its support of the Geothermal Energy and Geofluids (GEG.ethz.ch) Group at ETH Zurich, Switzerland.

We would like to thank our colleagues, Xiang-Zhao Kong and Isamu Naets, for all the helpful discussions. Furthermore, this work could not have been done without the visualization tool matplotlib (Hunter, 2007). We utilized the color palettes provided by Harrower and Brewer (2003); Davis (2019); Crameri (2021).





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
