# Peer review of "Validating the Nernst–Planck transport model under reaction-driven flow conditions using RetroPy v1.0"

_EGUsphere, 2022_

## Referee Comment (RC1)

**Referee's report on the paper**

**Validating the Nernst-Planck transport model under reaction-driven flow conditions using RetroPy v1.0**

Po-Wei Huang, Bernd Flemisch, Chao-Zhong Qin,
Martin O. Saar, and Anozie Ebigbo

The authors study the transport with electrodiffusion in continuous media, in the two-dimensional case, $2D$. They start with the continuity equations

$$\frac{\partial C_i}{\partial t} + \nabla \cdot J_i = 0, \quad i = 1, ..., N-1, \tag{1}$$

where $C_i$ mean concentrations and the fluxes $J_i$ are composed of three parts: Fick's part, Nernst-Planck's part an Darken's part as follows

$$J_i = -D_i \nabla C_i + \frac{z_i C_i D_i F}{RT} E + C_i u, \quad i = 1, ..., N-1. \tag{2}$$

Darken's velocity $u$ fulfills Darcy's law

$$u = -\frac{k}{\eta}(\nabla p - \rho g) \tag{3}$$

and the incompressibility condition holds

$$\nabla \cdot (\rho u) = 0. \tag{4}$$

Under the electroneutral condition

$$\sum_{i=1}^{N-1} z_i C_i = 0, \tag{5}$$

the system (1) leads to the stationary equation

$$-\nabla \cdot \left( \sum_{i=1}^{N-1} \frac{D_i C_i (z_i F)^2}{RT} E - \sum_{i=1}^{N-1} D_i z_i F \nabla C_i \right) = 0. \tag{6}$$

The authors postulate, by the paper due to Tabrizinejadas et al., 2021, that the electric field has the form

$$E = \frac{RT \sum_{i=1}^{N-1} d_i z i \nabla C_i}{F \sum_{i=1}^{N-1} (z_i)^2 D_k C_k}. \tag{7}$$

Here is a very big mistake! The formula (7) is true in the 1D case only, if for example $\sum_{i=1}^{N-1} z_i J_i = 0$ on the boundary of a domain. Then (7) is implied by (6) - see the paper:
1. Bernard P. Boudreau, Filip J.R. Meysman, Jack J. Middelburg, *Multicomponent ionic diffusion in porewaters: Coulombic effects revisited*, Earth and Planetary Science Letters 222 (2004), 653-666.

Tabrizinejadas et al., 2021 study the 1D, 2D and 3D models and they refer to the paper 1., so they are right in 1D only. I understand that the authors get some pictures, but mathematics has its laws.

In 2D and 3D we can for example assume that $E$ is an irrotational vector field, $\nabla \times E = 0$, and then $E$ is a potential field

$$E = -\nabla\varphi. \tag{8}$$

This equation together with (6) imply the Poisson equation on $\varphi$ of the form

$$\nabla \cdot \left( \sum_{i=1}^{N-1} \frac{D_i C_i (z_i F)^2}{RT} \nabla\varphi + \sum_{i=1}^{N-1} D_i z_i F \nabla C_i \right) = 0. \tag{9}$$

I refer the authors to the papers in which a similar situation appears, but with the drift $u$ instead of the electric field $E$:

2. B. Bożek, L. Sapa, K. Tkacz-Śmiech, M. Zajusz, M. Danielewski, *Compendium about multicomponent interdiffusion in two dimensions*, Metallurgical and Materials Transactions A 52A (2021), 3221-3231.

3. L. Sapa, B. Bożek, K. Tkacz-Śmiech, M. Zajusz, M. Danielewski, *Interdiffusion in many dimensions: mathematical models, numerical simulations and experiment*, Mathematics and Mechanics of Solids 25 (2020), 2178-2198.

4. B. Bożek, L. Sapa, M. Danielewski, *Difference methods to one and multidimensional interdiffusion models with Vegard rule*, Mathematical Modelling and Analysis 24 (2019), 276-296.

The paper has an engineering and numerical nature, and is interesting. But the error I mentioned above must be reliably described and explained, even if the authors are currently unable to do calculations in 2D and 3D with the equation (9). I suggest to start with experiments and calculations in 1D. Moreover, the jump operator $[\bullet]$ should be defined and it would be better to write $c_i$ instead of $C_i$. Domain dimension in experiments and calculations should be written in Abstract.

CONCLUSION
The paper need a major revision.

---

## Referee Comment (RC2)

Po-Wei Huang, Bernd Flemisch, Chao-Zhong Qin, Martin O. Saar, and Anozie Ebigbo:
*Validating the Nernst - Planck transport model under reaction-driven flow conditions using RetroPy v1.0*
* * *
The authors apply Nernst-Planck equation in order to model reactive transport processes in natural environment for single-phase multicomponent system. The importance of Nernst-Planck model is that it takes into account the electromigration of ionic species that have different diffusivities. On the other hand, if all species have the same diffusivity, the single-diffusivity model is obtained. For both models it is taken into account the electroneutrality condition and the zero-charge accumulation assumption.

The authors solve equations numerically. Mixed finite element method formulation in order to obtain fluid velocity and pressure is used. For the transport equation the authors have used finite volume scheme, in which they use upwinding for the advective flux, which is common approach in the equations for transport processes with convection and diffusion. For time-stepping they have used explicit scheme for advection term and Crank-Nicolson for the diffusion and the Nernst-Planck terms.

In this manuscript the results are presented for numerical experiments that deal with chemically driven convection of acid-base systems, and convective dissolution of $CO_2$ in reactive alkaline solutions. The authors compare the results of Nernst-Planck and single-diffusivity model mutually, and with experimental figures from the literature. Furthermore, in the last section authors discuss and explain on their obtained results.

I have found this manuscript very interesting. First numerical example is very interesting, and it provides the value of Nernst-Planck model. I recommend the manuscript for publication. The authors also provide a large amount of references about the topic of this manuscript. Furthermore all the simulations are obtained using RetroPy 1.0 that is available online.

In my opinion there are some minor points that are not clearly written:

○ In Figure 1, a flow chart is represented. I suggest to use the same notation for the velocity as in the text of the manuscript

○ Since the Nernst-Planck equation is non-stationary, for the sake of clarity, I suggest that you write the equation from which initial molar concentrations are obtained, to make everything mathematically more clearly, since for the boundary conditions no-flow boundary conditions are taken, the initial conditions must be written clearly.

○ Could you please elaborate, how large is the system in (32) when you update the velocity, and explain the reason why you have chosen to solve it with direct solver? (Just to comment: since you have used direct solver for linear system (in which

possibly positive-definite matrix is added ($rB^TB$)) maybe you could try to use some iterative solver from PETSc, the library that was also used in this work for resolving nolinearity in transport equation )

○ Could you please elaborate about the stepsizes $\Delta t$ that have used in your simulations?

---

## Author Comment (AC1)

Dear Professor Sapa,

We thank you for the efforts you took to review our paper and provide critical comments. We have carefully addressed the comments, especially in explaining the derivation of the electric field. The references you provided are relevant and helpful, and we have cited them. The domain dimensions of the experiment are included in the abstract, and the jump operator is defined. We hope the revisions meet your standards.

Below we provide the point-by-point responses. All modifications of the manuscript have been highlighted in red.

Sincerely,
Po-Wei Huang
powei.huang@erdw.ethz.ch
Postdoctoral Researcher, Geothermal Energy and Geofluids Group,
Institute of Geophysics, ETH Zurich

**Reviewer's comments**

**Comment 1** — ... leads to the stationary equation

$$-\nabla \cdot \left( \sum_{i=1}^{N-1} \frac{D_i C_i (z_i F)^2}{RT} \vec{E} - \sum_{i=1}^{N-1} D_i z_i F \nabla C_i \right) = 0 \,. \tag{6}$$

The authors postulate, by the paper due to Tabrizinejadas et al., 2021, that the electric field has the form

$$\vec{E} = \frac{RT \sum_{j=1}^{N-1} D_j z_j \nabla C_j}{F \sum_{k=1}^{N-1} (z_k)^2 D_k C_k} \,. \tag{7}$$

Here is a very big mistake! The formula (7) is true in the 1D case only, if for example $\sum_{i=1}^{N-1} z_i J_i = 0$ on the boundary of a domain. Then (7) is implied by (6) - see the paper:

1. Bernard P. Boudreau, Filip J.R. Meysman, Jack J. Middelburg, *Multicomponent ionic diffusion in porewaters: Coulombic effects revisited*, Earth and Planetary Science Letters 222 (2004), 653–666.

Tabrizinejadas et al., 2021 study the 1D, 2D and 3D models and they refer to the paper 1., so they are right in 1D only. I understand that the authors get some pictures, but mathematics has its laws.

**Reply**: We appreciate the reviewer pointing out this mistake. We agree that the divergence-free condition, Eq. (6), does not generally imply there are no fluxes, Eq. (7). We rewrite this part in section 2.1.4 to clarify that we employ the null-current condition to achieve strict local electroneutrality at all times. We also cited papers that utilized the null-current condition when modeling multicomponent transport. Please refer to the excerpts below.

**2.1.4 Modeling of the electrophoretic flux**

We assume the aqueous solution is locally electroneutral,

$$\sum_{i=1}^{N-1} z_i C_i = 0\,, \tag{14}$$

$$\frac{\partial \rho_{\mathrm{E}}}{\partial t} = 0\,.$$

 which is an approximation considering the Debye length (typically on the order of nanometers) vanishes at the length scale of the considered system (Dickinson et al., 2011). The electroneutrality assumption eliminates the barycentric flux term in the charge conservation equation. We consider Fickian diffusive flux, Eq. (6), and the electrophoretic flux, Eq. (10), so that the charge conservation is given by

$$\frac{\partial \rho_{\mathrm{E}}}{\partial t} = -\nabla \cdot \left( \sum_{i=1}^{N-1} \frac{D_i C_i (z_i F)^2}{RT} \boldsymbol{E} - \sum_{i=1}^{N-1} D_i z_i F \nabla C_i \right) = 0\,. \tag{15}$$

...

The model that combines mass conservation, Eq. (11), and  Poisson's equation of electrostatics, is known as the Poisson–Nernst–Planck (PNP) model (Pamukcu, 2009).

...

In this work, we enforce the null current condition

$$\sum_{i=1}^{N-1} \frac{D_i C_i (z_i F)^2}{RT} \boldsymbol{E} - \sum_{i=1}^{N-1} D_i z_i F \nabla C_i = 0 \tag{18}$$

to ensure that no charge accumulates at the continuum scale of interest,

$$\frac{\partial \rho_{\mathrm{E}}}{\partial t} = 0\,. \tag{19}$$

This results in a simplification of the PNP model. The null current condition has been utilized in modeling multicomponent ionic transport (Lichtner, 1985; Cappellen and Gaillard, 1996; Giambalvo et al., 2002; Muniruzzaman and Rolle, 2019; Cogorno et al., 2022; López-Vizcaíno et al., 2022). The combined assumptions of local electroneutrality and null current are referred to as strict electroneutrality by Lees et al., 2017. Furthermore, If there are no sources of the electric field (Tabrizinejadas et al., 2021), then the electric field can be represented by

$$\boldsymbol{E} = \frac{RT}{F} \frac{\sum_{j=1}^{N-1} D_j z_j \nabla C_j}{\sum_{k=1}^{N-1} (z_k)^2 D_k C_k}\,, \tag{20}$$

where we use $j$ and $k$ as summation indices.

**Comment 2** — In 2D and 3D we can for example assume that $\vec{E}$ is an irrotational vector field, $\nabla \times \vec{E} = 0$, and then $\vec{E}$ is a potential field

$$\vec{E} = -\nabla\varphi. \tag{8}$$

This equation together with (6) imply the Poisson equation on $\varphi$ of the form

$$\nabla \cdot \left( \sum_{i=1}^{N-1} \frac{D_i C_i (z_i F)^2}{RT} \nabla\varphi + \sum_{i=1}^{N-1} D_i z_i F \nabla C_i \right) = 0. \tag{9}$$

...(see Comment 3) The paper has an engineering and numerical nature, and is interesting. But the error I mentioned above must be reliably described and explained, even if the authors are currently unable to do calculations in 2D and 3D with the equation (9). I suggest to start with experiments and calculations in 1D.

**Reply**: Thank you for considering our paper interesting! In this work, we do not consider solving the electric potential, $\varphi$, because we utilized the null current assumption, and there is no imposed electric potential in the experiments we compared. We have explained and edited the derivations by utilizing the null current assumption, see the excerpts in Comment 1. Numerical benchmarks of multicomponent diffusion using the null current assumption in 1D and 2D are performed by Rasouli et al., 2015.

Although not explicitly stated in the paper, we did numerical tests of multicomponent diffusion of two species in 1D. The tests are named 'dg0_charge_balance_test.py' and 'dg0_exp_charge_balance_test.py', and they are located in the 'tests' folder.

**Comment 3** — I refer the authors to the papers in which a similar situation appears, but with the drift $u$ instead of the electric field $\vec{E}$:

2. B. Bożek, L. Sapa, K. Tkacz-Śmiech, M. Zajusz, M. Danielewski, *Compendium about multicomponent interdiffusion in two dimensions*, Metallurgical and Materials Transactions A 52A (2021), 3221–3231.

3. L. Sapa, B. Bożek, K. Tkacz-Śmiech, M. Zajusz, M. Danielewski, *Interdiffusion in many dimensions: mathematical models, numerical simulations and experiment*, Mathematics and Mechanics of Solids 25 (2020), 2178–2198.

4. B. Bożek, L. Sapa, M. Danielewski, *Difference methods to one and multidimensional interdiffusion models with Vegard rule*, Mathematical Modelling and Analysis 24 (2019), 276–296.

**Reply**: We have read the suggested work; they are helpful in addressing Comment 1. We consider the models of multicomponent interdiffusion between solids relevant to our work and have cited them in the paragraph where we discuss the mathematical properties of the PNP model.
* * *
**2.1.4 Modeling of the electrophoretic flux**

The PNP model aims at resolving both the electric potential and the molar concentrations, subject to the boundary conditions of the electric potential. ... Models similar to the PNP model also arise in modeling multicomponent interdiffusion of solids (Bożek et al., 2019), where the experimental comparison and the development of numerical methods are studied by Sapa et al., 2020 and Bożek et al., 2021.

**Comment 4** — Moreover, the jump operator [•] should be defined.

**Reply**: We apologize for being unclear. We have made the following modifications to clarify the meaning of the jump operator. Furthermore, we specified the software packages we used to define the variational formulations.
* * *
**2.2.2 Transport of fluid components**

where $\Gamma_{\text{int}}$ denotes the interior  cell interfaces, [•] is the jump operator that evaluates the difference of a function across a common interface of two cells, and $h$ is the distance between the cell centers. ... UFL (Alnæs, 2012) is utilized for defining the variational forms in our code implementation.
* * *
**Comment 5** — It would be better to write $c_i$ instead of $C_i$.

**Reply**: We have employed consistent definitions of our variables to make sure everything is clear. No change is made.

**Comment 6** — Domain dimension in experiments and calculations should be written in Abstract.

**Reply**: Thank you for the comment! We think it is helpful to state the dimensions of the experiments in the abstract. Please refer to the modifications below.
* * *
**Abstract**

... To demonstrate the advantages of the Nernst–Planck model, we compare the simulation results of transport under reaction-driven flow conditions using the Nernst–Planck model with those of the commonly used single-diffusivity model. All simulations are also compared to well-defined experiments on the scale of centimeters. ...
* * *
**References**

Alnæs, Martin Sandve (2012). "UFL: a finite element form language". In: *Automated Solution of Differential Equations by the Finite Element Method: The FEniCS Book*. Ed. by Anders Logg, Kent-Andre Mardal, and Garth N Wells. Berlin, Heidelberg: Springer Berlin Heidelberg, pp. 303–338. ISBN: 978-3-642-23099-8. DOI: 10.1007/978-3-642-23099-8_17.

Bożek, Bogusław, Lucjan Sapa, and Marek Danielewski (2019). "Difference Methods to One and Multidimensional Interdiffusion Models with Vegard Rule". In: *Math. Model. Anal.* 24 (2), pp. 276–296. DOI: 10.3846/mma.2019.018.

Bożek, Bogusław et al. (2021). "Compendium About Multicomponent Interdiffusion in Two Dimensions". In: *Metall. Mater. Trans. A* 52.8, pp. 3221–3231. DOI: 10.1007/s11661-021-06267-9.

Cappellen, Philippe Van and Jean-Francois Gaillard (1996). "Chapter 8. Biogeochemical Dynamics in Aquatic Sediments". In: *Reactive Transport in Porous Media*. Ed. by Peter C. Lichtner, Carl I. Steefel, and Eric H. Oelkers. De Gruyter, pp. 335–376. ISBN: 9781501509797. DOI: 10.1515/9781501509797-011.

Cogorno, Jacopo et al. (2022). "Dimensionality effects on multicomponent ionic transport and surface complexation in porous media". In: *Geochim. Cosmochim. Ac.* 318, pp. 230–246. ISSN: 0016-7037. DOI: 10.1016/j.gca.2021.11.037.

Dickinson, E. J. F., J. G. Limon-Peterson, and R. G. Compton (2011). "The electroneutrality assumption in electrochemistry". In: *J. Solid State Electr.* 15, pp. 1335–1345. DOI: 10.1007/s10008-011-1323-x.

Giambalvo, Emily R. et al. (2002). "Effect of fluid-sediment reaction on hydrothermal fluxes of major elements, eastern flank of the Juan de Fuca Ridge". In: *Geochim. Cosmochim. Ac.* 66.10, pp. 1739–1757. ISSN: 0016-7037. DOI: 10.1016/S0016-7037(01)00878-X.

Lees, Eitan et al. (2017). "The electroneutrality constraint in nonlocal models". In: *J. Chem. Phys.* 147.12, p. 124102. DOI: 10.1063/1.5003915.

Lichtner, Peter C. (1985). "Continuum model for simultaneous chemical reactions and mass transport in hydrothermal systems". In: *Geochim. Cosmochim. Ac.* 49.3, pp. 779–800. ISSN: 0016-7037. DOI: 10.1016/0016-7037(85)90172-3.

López-Vizcaíno, Rubén et al. (2022). "A modeling approach for electrokinetic transport in double-porosity media". In: *Electrochim. Acta* 431, p. 141139. ISSN: 0013-4686. DOI: 10.1016/j.electacta.2022.141139.

Muniruzzaman, Muhammad and Massimo Rolle (2019). "Multicomponent Ionic Transport Modeling in Physically and Electrostatically Heterogeneous Porous Media With PhreeqcRM Coupling for Geochemical Reactions". In: *Water Resour. Res.* 55.12, pp. 11121–11143. DOI: 10.1029/2019WR026373.

Pamukcu, Sibel (2009). "Electrochemical Transport and Transformations". In: *Electrochemical Remediation Technologies for Polluted Soils, Sediments and Groundwater*. John Wiley & Sons, Ltd. Chap. 2, pp. 29–64. ISBN: 9780470523650. DOI: 10.1002/9780470523650.ch2.

Rasouli, Pejman et al. (2015). "Benchmarks for multicomponent diffusion and electrochemical migration". In: *Computat. Geosci.* 19.3, pp. 523–533. DOI: 10.1007/s10596-015-9481-z.

Sapa, Lucjan et al. (2020). "Interdiffusion in many dimensions: mathematical models, numerical simulations and experiment". In: *Math. Mech. Solids* 25.12, pp. 2178–2198. DOI: 10.1177/1081286520923376.

Tabrizinejadas, Sara et al. (2021). "On the Validity of the Null Current Assumption for Modeling Sorptive Reactive Transport and Electro-Diffusion in Porous Media". In: *Water* 13.16. ISSN: 2073-4441. DOI: 10.3390/w13162221.

---

## Author Comment (AC2)

Dear Reviewer,

We thank you for reviewing our paper and giving us encouraging comments. We took all your comments into consideration, particularly in clarifying the initial molar concentrations of all simulations.

Below we provide the point-by-point responses. All modifications of the manuscript have been highlighted in red.

Sincerely,
Po-Wei Huang
powei.huang@erdw.ethz.ch
Postdoctoral Researcher, Geothermal Energy and Geofluids Group,
Institute of Geophysics, ETH Zurich

**Reviewer's comments**

**Comment 1** — In Figure 1, a flow chart is represented. I suggest to use the same notation for the velocity as in the text of the manuscript.

**Reply**: Thank you for pointing out the inconsistent notation in the flow chart. Throughout the manuscript, we used $u$ as the notation for the barycentric velocity. However, in Figure 1, we used $\vec{v}$ to represent the barycentric velocity. We corrected the flow chart. Please refer to the excerpts below.

[Figure]

**Figure 1.** Flow chart of the simulation procedures

**Comment 2** — Since the Nernst–Planck equation is non-stationary, for the sake of clarity, I suggest that you write the equation from which initial molar concentrations are obtained, to make everything mathematically more clearly, since for the boundary conditions no-flow boundary conditions are taken, the initial conditions must be written clearly.

**Reply**: We clarify the initial conditions of the mass transport equations by adding Table 3 to Section 2.3. Whenever the setup of the simulations is discussed in the text, we add a sentence that refers to Table 3 to ensure the initial conditions are clarified. The table is miniaturized to fit in the excerpts below. Please refer to the corrected manuscript for the full-sized table.

**2.3 Selected experiments for model evaluation**

No-flow boundary conditions are prescribed on all sides of the simulation domain. Table 3 specifies the chemical compositions of the aqueous solutions, which defines the initial conditions of the mass transport equations.

Table 3. The chemical compositions of the aqueous solutions. The unit of the chemical compositions is in $mol\,L^{-1}$. In the table, dashes denote "not applicable" because the chemical species are not considered in the simulation cases.

| | $H^+$ | $OH^-$ | $Na^+$ | $Cl^-$ | $Li^+$ | $NO_3^-$ | $HCO_3^-$ | $CO_3^{2-}$ | $CO_2(aq)$ | $H_2O(l)$ |
|---|---|---|---|---|---|---|---|---|---|---|
| 1M HCl | 1.0 | $2.109 \times 10^{-14}$ | $10^{-20}$ | 1.0 | - | - | - | - | - | 54.170 |
| 1M NaOH | $2.199 \times 10^{-14}$ | 1.0 | 1.0 | $10^{-20}$ | - | - | - | - | - | 55.360 |
| 1.4M NaOH | $1.537 \times 10^{-14}$ | 1.4 | 1.4 | - | - | $10^{-20}$ | - | - | - | 55.361 |
| 1.5M $HNO_3$ | 1.5 | $1.287 \times 10^{-14}$ | $10^{-20}$ | - | - | 1.5 | - | - | - | 52.712 |
| 0.01 M LiOH | $1.235 \times 10^{-12}$ | 0.01 | - | - | 0.01 | - | $10^{-20}$ | $10^{-20}$ | $10^{-20}$ | 55.345 |

**2.3.1 Chemically driven convection of acid–base systems**

Please refer to the left panel of Figure 2 for the setup. Table 3 lists the chemical composition of 1 M HCl and 1 M NaOH. The prescribed minimum timestep size is $3 \times 10^{-2}$ seconds, and the maximum timestep size is 2.0 seconds.
...
The setup is shown in Figure 2, and Table 3 lists the initial chemical composition of 1.5 M $HNO_3$ and 1.4 M NaOH. The prescribed minimum timestep size is $1 \times 10^{-3}$ seconds, and the maximum timestep size is 1.0 seconds.

**2.3.2 Convective dissolution of CO2 in reactive alkaline solutions**

We consider the following species as the main fluid components in the case of convective dissolution of $CO_2$: $H^+$, $OH^-$, $Li^+$, $CO_2(aq)$, $HCO_3^-$, $CO_3^{2-}$, and $H_2O(l)$, and the initial conditions are presented in Table 3. The prescribed minimum and maximum timestep sizes are $5 \times 10^{-3}$ seconds and 1.5 seconds, respectively.

**Comment 3** — Could you please elaborate, how large is the system in (32) when you update the velocity, and explain the reason why you have chosen to solve it with direct solver? (Just to comment: since you have used direct solver for linear system (in which possibly positive-definite matrix is added $(rB^T B)$) maybe you could try to use some iterative solver from PETSc, the library that was also used in this work for resolving nolinearity in transport equation )

**Reply**: Since we use the Raviart–Thomas basis for the velocity, the number of degree of freedom is the same as the number of cell faces (edges for 2D meshes). For the simulation cases (HCl–NaOH, $HNO_3$–NaOH, $CO_2$ dissolution), the edge counts are 52053, 17464, and 138576, respectively. We added the range of degrees of freedom into the manuscript.

We tried to use iterative solvers such as PETSc to calculate the first iteration of the augmented Lagrangian Uzawa's method, Eq. (32). We encountered convergence issues and could not find a suitable algebraic preconditioner to achieve convergence. This is why we calculate Eq. (32) by a direct solver.

To elaborate on why we did not calculate Eq. (32) by an iterative solver, we referenced Fortin and Glowinski, 1983 in Section 2.2.1 to motivate the use of direct solvers. Please refer to the excerpts below.

**2.2.1 The barycentric flux**

However, the matrix $A + rB^T B$ becomes more ill-conditioned as $r$ increases, which may lead to a large number of iterations to solve when applying an iterative method (Fortin and Glowinski, 1983). Hence, we use the MUMPS (Amestoy et al., 2001; Amestoy et al., 2019) direct solver to update the velocity, Eq. (32). The degrees of freedom of $\boldsymbol{u}$ in our simulations range from $17\,000$ to $140\,000$.

**Comment 4** — Could you please elaborate the stepsizes $\Delta t$ that have used in your simulations?

**Reply**: We added descriptions of the adaptive timestep sizes in Section 2.2.2. Furthermore, we added detailed timestep sizes $\Delta t$ we used in the simulations in Sections 2.3.1 and 2.3.2. Please refer to the excerpts below and the excerpts in Comment 2.

**2.2.2 Transport of fluid components**

With regard to the time-stepping schemes, we use an explicit scheme for the upwind advection term and the Crank–Nicolson scheme for the diffusion and Nernst–Planck terms. The timestep size $\Delta t$ is determined in an adaptive manner, where the minimum and maximum timestep sizes are prescribed for each simulation.

**References**

Amestoy, P. R. et al. (2001). "A Fully Asynchronous Multifrontal Solver Using Distributed Dynamic Scheduling". In: *SIAM J. Matrix Anal. A.* 23.1, pp. 15–41. DOI: 10.1137/S0895479899358194.

Amestoy, P. R. et al. (2019). "Performance and Scalability of the Block Low-Rank Multifrontal Factorization on Multicore Architectures". In: *ACM T. Math. Software* 45 (1), pp. 1–26. DOI: 10.1145/3242094.

Fortin, M. and R. Glowinski (1983). "Augmented Lagrangian Methods in Quadratic Programming". In: *Augmented Lagrangian Methods: Applications to the Numerical Solution of Boundary-Value Problems.* Ed. by Michel Fortin and Roland Glowinski. Vol. 15. Studies in Mathematics and Its Applications. Elsevier. Chap. 1, pp. 1–46. DOI: 10.1016/S0168-2024(08)70026-2.

---

## Author Response (AR1)

Dear Editor,

We have provided point-by-point responses to the reviewer's comments during open discussion. We assessed the reviewer's comments and revised the manuscript. In addition to addressing the reviewer's comments, we corrected a typo in the main text and some bibliography issues (omitted book titles). Furthermore, we added references to the software we used, namely FEniCS and United Form Language (UFL).

In the track-changes file, all additions in the manuscript have been highlighted in red, and exclusions are crossed out.

Sincerely,
Po-Wei Huang
powei.huang@erdw.ethz.ch
Postdoctoral Researcher, Geothermal Energy and Geofluids Group,
Institute of Geophysics, ETH Zurich

---

## Referee Report (RR1)

**Second referee's report on the paper**

**Validating the Nernst-Planck transport model under reaction-driven flow conditions using RetroPy v1.0**

Po-Wei Huang, Bernd Flemisch, Chao-Zhong Qin,
Martin O. Saar, and Anozie Ebigbo

In a new version, the Authors have taken into account all my remarks. Obviously (18) implies (19). But the representation (20) is a postulate only. The Authors added the appropriate sentence "This result is a simplification of the PNP model". By the way, it should be written "This result is..." instead of "This results in...". It would be interesting from a mathematical point of view to keep (6) (see my first referee's report) with some "tricks" on $E$. I know that it can be very difficult.

CONCLUSION
The paper can be published in Egusphere.